# MULTI-MODE DEEP MATRIX AND TENSOR FACTORIZATION

**Jicong Fan**[1,2]
[1]School of Data Science, The Chinese University of Hong Kong (Shenzhen), China
[2]Shenzhen Research Institute of Big Data, China
`fanjicong@cuhk.edu.cn`

## ABSTRACT

Recently, deep linear and nonlinear matrix factorizations gain increasing attention in the area of machine learning. Existing deep nonlinear matrix factorization methods can only exploit partial nonlinearity of the data and are not effective in handling matrices of which the number of rows is comparable to the number of columns. On the other hand, there is still a gap between deep learning and tensor decomposition. This paper presents a framework of multi-mode deep matrix and tensor factorizations to explore and exploit the full nonlinearity of the data in matrices and tensors. We use the factorization methods to solve matrix and tensor completion problems and prove that our methods have tighter generalization error bounds than conventional matrix and tensor factorization methods. The experiments on synthetic data and real datasets showed that the proposed methods have much higher recovery accuracy than many baselines.

## 1 INTRODUCTION

Low-rank matrices and tensors are pervasive in sciences and engineering. Low-rank matrix completion (LRMC) (Srebro & Shraibman, 2005; Candès & Recht, 2009; Recht, 2011; Hu et al., 2013; Hardt, 2014; Chen et al., 2014; Shamir & Shalev-Shwartz, 2014; Sun & Luo, 2016; Fan & Chow, 2017; Fan et al., 2020; Kümmerle & Sigl, 2018; Fan et al., 2019) and low-rank tensor completion (LRTC) (Gandy et al., 2011; Acar et al., 2011; Liu et al., 2012; Kressner et al., 2014; Yuan & Zhang, 2016; Foster & Risteski, 2019), as shown in Figure 1, aim to recover the missing entries of a low-rank matrix or tensor. LRMC and LRTC are very usefully in data preprocessing, image and video inpainting, and collaborative filtering (Fan & Cheng, 2018; Liu et al., 2012). For instance, in collaborative filtering (e.g. the recommendation problem of Netflix), the rating matrix given by users on items is often highly incomplete because each user can only rate a few items and each item is often rated by a few users. If the unknown entries of the rating matrix are predicted, recommendation can then be made, accordingly, for users or items. Since rating matrices often have potentially low-rank structures, we can use LRMC or even LRTC to recover the missing entries for recommendation.

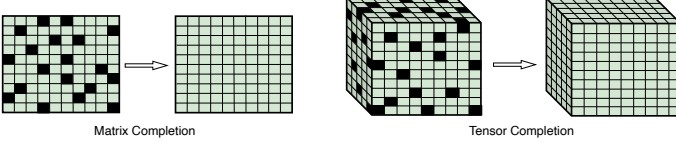

Matrix Completion        Tensor Completion

Figure 1: Intuitive examples of matrix and tensor completion (black square indicates missing value).

**Notation** Throughout the paper, '$x$', '$\boldsymbol{x}$', '$\boldsymbol{X}$', and '$\boldsymbol{\mathcal{X}}$' denote scalar, column vector, matrix, and tensor respectively. '$\boldsymbol{X}^{(j)}$' (or '$\boldsymbol{X}_j$') denotes a matrix with index $j$. $\|\cdot\|_F$ denotes the Frobenius norm of matrix or tensor. $\|\cdot\|_*$ denotes the nuclear norm of matrix, i.e. sum of the singular values. $\|\cdot\|_\infty$ denotes the maximum absolute element in a matrix or tensor. '$\boldsymbol{\mathcal{C}} \times_j \boldsymbol{A}$' denotes the $j$-mode product of a tensor $\boldsymbol{\mathcal{C}}$ with a matrix $\boldsymbol{A}$. $\mathcal{P}_\Omega(\boldsymbol{X})$ denotes the projection onto the set $\Omega$, i.e., $[\mathcal{P}_\Omega(\boldsymbol{X})]_{ij} = [\boldsymbol{X}]_{ij}$ if $(i, j) \in \Omega$ and $[\mathcal{P}_\Omega(\boldsymbol{X})]_{ij} = 0$ otherwise. $|\Omega|$ denotes the cardinality of $\Omega$. $g(\cdot)$ denotes an activation function and $h_l$ denotes the number of units in layer $l$ of a neural network.

**LRMC** In the past decade, a lot of algorithms with theoretical guarantee or/and empirical success have been proposed for LRMC. For example, Candès & Recht (2009) proved that a low-rank matrix can be recovered exactly with high probability from a few entries sampled uniformly at random, via nuclear norm minimization. Nie et al. (2012) proposed to minimize the Schatten-$p$ quasi-norm for LRMC. Nuclear norm and Schatten-$p$ quasi-norm minimizations are based on singular value decomposition and hence have high computational cost when the size of the matrix is very large. In addition, they do not incorporate the rank prior that are possibly available in practice, though could be inaccurate. In contrast, low-rank factorization based methods (Srebro & Shraibman, 2005; Srebro & Salakhutdinov, 2010; Wen et al., 2012; Hardt, 2014; Sun & Luo, 2016; Jin et al., 2016; Shang et al., 2016) are scalable to big matrices and can provide high recovery accuracy if the factorization size is properly determined. In fact, regularized matrix factorizations are closely related to nuclear norm and Schatten-$p$ quasi-norm minimizations. For example, it is known (Srebro et al., 2005; Rennie & Srebro, 2005) that $\|\boldsymbol{X}\|_* = \min_{\boldsymbol{AB}=\boldsymbol{X}} \|\boldsymbol{A}\|_F \|\boldsymbol{B}\|_F = \min_{\boldsymbol{AB}=\boldsymbol{X}} \frac{1}{2} \left( \|\boldsymbol{A}\|_F^2 + \|\boldsymbol{B}\|_F^2 \right)$. For LRMC, we can solve

$$\underset{\boldsymbol{A},\boldsymbol{B}}{\text{minimize}} \ \frac{1}{2} \|\mathcal{P}_\Omega(\boldsymbol{Y} - \boldsymbol{AB})\|_F^2 + \frac{\lambda}{2} \left( \|\boldsymbol{A}\|_F^2 + \|\boldsymbol{B}\|_F^2 \right), \tag{1}$$

where $\boldsymbol{A} \in \mathbb{R}^{m \times d}$, $\boldsymbol{B} \in \mathbb{R}^{d \times n}$, and $0 < d \leq \min\{m, n\}$. Interestingly, Gunasekar et al. (2017) found and proved that gradient descent for (1) with $\lambda = 0$ and $d = m = n$ converges to the minimum nuclear norm solution, provided that the learning rate is small enough and the initializations are close enough to the origin. Fan et al. (2019) proposed a class of rank regularizers called factor group sparse regularizer (FGSR) as the variational forms for Schatten-$p$ quasi-norms, e.g.,

$$\min_{\boldsymbol{X}=\boldsymbol{AB}} \frac{1}{q} \|\boldsymbol{A}\|_{2,q}^q + \alpha \|\boldsymbol{B}\|_{2,1} = (1 + 1/q) \alpha^{q/(q+1)} \|\boldsymbol{X}\|_{S_{q/(q+1)}}^{q/(q+1)}, \tag{2}$$

where $\alpha > 0$ and $q \in \{1, \frac{1}{2}, \frac{1}{4}, \ldots\}$. When $q = 1$, one gets Schatten-1/2 quasi-norm. Later, Giampouras et al. (2020) extended the variational form to arbitrary $p \in (0, 1]$.

**LRTC** A few LRMC methods have been directly extended to LRTC (Acar et al., 2011; Liu et al., 2012; Kressner et al., 2014). One may categorize LRTC algorithms according to the different tensor factorization models such as CP (CANDECOMP/PARAFAC) decomposition (Kolda & Bader, 2009; Jain & Oh, 2014), Tucker decomposition (Xu et al., 2013; Xie et al., 2018), tensor singular value decomposition (Kilmer et al., 2013; Zhang & Aeron, 2016), and tensor ring decomposition (Zhao et al., 2016; Huang et al., 2020). Take the CP decomposition based LRTC as an example, one can just learn a CP decomposition from the partial observations (Acar et al., 2011). Yuan & Zhang (2016) provided the sample complexity for the exact recovery of LRTC via nuclear norm minimization, though the optimization is intractable because computing the nuclear norm of a tensor is NP-hard (Hillar & Lim, 2013). Barak & Moitra (2016) and Foster & Risteski (2019) exploited the sum-of-squares hierarchy for LRTC and provided theoretical guarantees for the recovery. Razin et al. (2021) studied the implicit regularization in tensor factorization and proved that gradient descent with small learning rate and near-zero initialization gives rise to a bias towards low tensor rank solutions. Liu et al. (2019) introduced convolutional neural network to tensor completion. The method can model the possibly complex interactions inside tensors while preserving the desired low-rank structure.

To complete a third-order (w.l.g) tensor based on the Tucker decomposition, we may solve

$$\underset{\boldsymbol{\mathcal{X}},\boldsymbol{A}_1,\boldsymbol{A}_2,\boldsymbol{A}_3,\boldsymbol{\mathcal{C}}}{\text{minimize}} \ \frac{1}{2} \|\boldsymbol{\mathcal{X}} - \boldsymbol{\mathcal{C}} \times_1 \boldsymbol{A}_1 \times_2 \boldsymbol{A}_2 \times_3 \boldsymbol{A}_3\|_F^2, \quad \text{subject to } \mathcal{P}_\Omega(\boldsymbol{\mathcal{X}}) = \mathcal{P}_\Omega(\boldsymbol{\mathcal{Y}}), \tag{3}$$

where we can also consider regularization or constraint on the factors $\boldsymbol{A}_1$, $\boldsymbol{A}_2$, $\boldsymbol{A}_3$, and $\boldsymbol{\mathcal{C}}$. For instance, Xu et al. (2013) reformulated (3) to all-mode matricization and solved the problem by alternating minimization. Xie et al. (2018) proposed to minimize the number of nonzero elements of $\boldsymbol{\mathcal{C}}$ and the ranks of the matricizations of $\boldsymbol{\mathcal{X}}$, in addition to (3).

**Deep Matrix Factorization** Recently, deep matrix factorization (DMF) (Fan & Cheng, 2018; Arora et al., 2019) gains increasing attention in machine learning. There are two types of DMF methods: linear DMF (Trigeorgis et al., 2016; Zhao et al., 2017; Arora et al., 2019) and nonlinear DMF (Xue et al., 2017; Wang et al., 2017; Fan & Cheng, 2018). A general formulation is

$$\boldsymbol{X} \approx g_1(\boldsymbol{A}_1 g_2(\boldsymbol{A}_2 \cdots g_{L-1}(\boldsymbol{A}_{L-1}\boldsymbol{A}_L)\cdots)), \tag{4}$$

where $\{g_i\}_{l=1}^{L-1}$ are activation functions. When all $g_i$ are linear, (4) is linear DMF. Otherwise, (4) is nonlinear DMF. Arora et al. (2019) found that adding depth to a linear matrix factorization can enhance an implicit tendency towards low-rank and can provide higher recovery accuracy than depth-2 factorization, though the implicit regularization in linear DMF may not be captured using simple mathematical norms. In (Fan & Cheng, 2018), the numerical results showed that nonlinear DMF is able to recover (with high accuracy) the missing entries of a full-rank matrix of which the columns are generated by a nonlinear low-dimensional latent variable model.

**Contributions of this paper**   This paper first provides theoretical analysis for why and when nonlinear DMF outperforms linear DMF in matrix completion. Second, this paper proposes a new matrix factorization method called two-mode nonlinear DMF. The method can exploit the full nonlinearity of the data while classical nonlinear DMF uses only partial nonlinearity of the data. Third, the paper presents a new tensor decomposition method called multi-mode nonlinear deep tensor factorization, which can be regarded as a high-order generalization of the two-mode nonlinear DMF. Finally, the paper shows that the multi-mode deep matrix and tensor factorization methods have tighter generalization error bounds than conventional factorization methods in matrix and tensor completion. Extensive numerical results verified the effectiveness of the proposed methods.

## 2   WHY AND WHEN DOES NONLINEAR DMF OUTPERFORM MF?

Nonlinear DMF methods can outperform linear MF methods empirically in many real tasks such as collaborative filtering (Xue et al., 2017; Fan & Cheng, 2018), image inpainting (Fan & Cheng, 2018), and disease modeling (Wang et al., 2017). The fundamental reason is that the data matrices in these tasks indeed have some nonlinear structures that cannot be exploited by linear MF methods. Nevertheless, in nonlinear DMF, the theoretical guarantee is very limited. In contrast, linear MF or even linear DMF have been well studied and guaranteed with sample complexity (Hardt, 2014; Sun & Luo, 2016; Jin et al., 2016), generalization bound (Srebro & Shraibman, 2005; Fan et al., 2019), or convergence (Gunasekar et al., 2017; Arora et al., 2019). In this section, we will provide some theoretical guarantee for nonlinear DMF in matrix completion.

**Assumption 1.** *Suppose* $\boldsymbol{Y} = \boldsymbol{X} + \boldsymbol{E}$*, where* $\boldsymbol{Y} \in \mathbb{R}^{n_1 \times n_2}$*. The columns of* $\boldsymbol{X}$ *are generated by*

$$\boldsymbol{x} = f(\boldsymbol{z}),$$

*where* $f : \mathbb{R}^d \to \mathbb{R}^{n_1}$ *denotes a nonlinear mapping,* $d < \min(n_1, n_2)$*, and* $\boldsymbol{z} \in \mathbb{R}^d$ *denotes a latent variable. The entries of the noise matrix* $\boldsymbol{E}$ *are drawn from* $\mathcal{N}(0, \sigma^2)$*.*

Suppose we observed a few entries of $\boldsymbol{Y}$ randomly (sampling without replacement) and want to recover $\boldsymbol{X}$ from the incomplete $\boldsymbol{Y}$. Consider the following method (Fan & Cheng, 2018):

$$\underset{\boldsymbol{W}_1,\dots,\boldsymbol{W}_L,\boldsymbol{Z}}{\text{minimize}} \frac{1}{2}\|\mathcal{P}_\Omega(\boldsymbol{Y} - \boldsymbol{W}_L g(\boldsymbol{W}_{L-1}\cdots g(\boldsymbol{W}_1\boldsymbol{Z})\cdots))\|_F^2 + \frac{\lambda_w}{2}\sum_{l=1}^{L}\|\boldsymbol{W}_l\|_F^2 + \frac{\lambda_z}{2}\|\boldsymbol{Z}\|_F^2, \quad (5)$$

where $\boldsymbol{Z} \in \mathbb{R}^{d \times n_2}$, $\boldsymbol{W}_l \in \mathbb{R}^{h_l \times h_{l-1}}$, $l = 1,\dots,L$, $h_L = n_1$, $h_0 = d$, and $h_{-1} = n_2$. $\lambda_w$ and $\lambda_z$ are regularization parameters. Here we have let all activation functions be the same without loss of generality and omitted the bias terms of the neural network for simplicity. According to the universal approximation theorems (Cybenko, 1989; Pinkus, 1999; Lu et al., 2017; Hanin & Sellke, 2017), $f$ can be well approximated by the neural network provided that $L$ is sufficiently large or at least one of $h_1,\dots,h_{L-1}$ is sufficiently large. The following theorem provides the excess risk bound for (5).

**Theorem 1.** *Suppose* $\boldsymbol{Z}$ *and* $\{\boldsymbol{W}_l\}_{l=1}^{L}$ *are given by* (5)*. Let* $\hat{\boldsymbol{X}} = \boldsymbol{W}_L g(\boldsymbol{W}_{L-1}\cdots g(\boldsymbol{W}_1\boldsymbol{Z})\cdots)$*. Suppose* $\|\boldsymbol{Z}\|_F \prod_{l=1}^{L}\|\boldsymbol{W}_l\|_F \leq \kappa$*,* $\max(\|\boldsymbol{Y}\|_\infty, \|\hat{\boldsymbol{X}}\|_\infty) \leq \xi$*, the Lipschitz constant of* $g$ *is* $\eta$*. Let* $N = n_1 n_2$*. Then with probability at least* $1 - 2N^{-1}$*, there exists a numerical constant* $c$ *such that*

$$\frac{1}{\sqrt{N}}\|\boldsymbol{X} - \hat{\boldsymbol{X}}\|_F \leq \frac{1}{\sqrt{|\Omega|}}\|\mathcal{P}_\Omega(\boldsymbol{Y} - \hat{\boldsymbol{X}})\|_F + \frac{1}{\sqrt{N}}\|\boldsymbol{E}\|_F + c\xi\left(\frac{\sum_{l=0}^{L}h_l h_{l-1}\log(\eta^{L-1}\xi^{-1}\kappa)}{|\Omega|}\right)^{1/4}.$$

In Theorem 1, the last term of the RHS of the inequality is equivalent to

$$\psi_1 = c\xi\big(|\Omega|^{-1}(n_2 d + h_{L-1}n_1 + \sum_{l=1}^{L-1}h_l h_{l-1})\log(\eta^{L-1}\xi^{-1}\kappa)\big)^{1/4}. \quad (6)$$

Most major activation functions (e.g. ReLU and sigmoid) are, at worst, 1-Lipschitz with respect to $\ell_2$ norm, which indicates that $\eta^{L-1}$ is not large. When $n_2$ is large enough, $\psi_1$ is linear with the latent dimension $d$ of the columns of $X$ when using (5). In practice, it is difficult to know the true $d$ in advance. But the Frobenius norm regularizations on $Z$ and $W_1$ are able to reduce the nuclear norm or even the rank of $W_1 Z$ theoretically, which means the $d$ in formulation (5) and Theorem 1 can be larger than the $d$ (ground truth) in Assumption 1.

Note that classical MF (depth-1) is a special case of (5) when $L = 1$ and $g$ is linear. However, because of the nonlinearity in the data generating model of Assumption 1, the rank of $X$ can be much higher than the intrinsic dimension of the data, i.e. $r := \text{rank}(X) \gg d$. For example, if $f$ is a $q$-order polynomial function, $r$ can be as large as $\binom{d+q}{q}$. For classical MF, the last term of the RHS of the inequality in Theorem 1 reduces to

$$\psi_2 = c\xi\big(|\Omega|^{-1}(n_1 r + r n_2)\log(\xi^{-1}\kappa')\big)^{1/4}. \tag{7}$$

Here $\kappa'$ may be less than the $\kappa$ in (6) but they are in the log operator. Suppose all the Frobenius norms are the same (the worst case, may not happen because the norms of the factors in a deep factorization are usually smaller than the norms of the factors in a shallow factorization), denoted by $\beta$. Then $\kappa = \beta^{L+1}$ and $\kappa' = \beta^2$, but $r \gg d$. In the experiments, we observed that $\kappa \approx (\kappa')^{1.3}$ when $L = 2$ and $\kappa \approx (\kappa')^{1.7}$ when $L = 3$, which are much better than the worst cases. Now comparing (7) with (6), we conclude that the nonlinear DMF provides a tighter generalization bound than the classical MF when $n_2$ is much larger than $n_1$ and $\sum_{l=1}^{L-1} h_l h_{l-1}$ is not too large. The second condition is realistic because we usually let $h_1, \ldots, h_{L-1} < n_1$ or we can just assume $h_1, \ldots, h_{L-1} < r$.

Consider the case that $n_2$ is not large compared to $n_1$, e.g. $n_2 = n_1$. In (6), suppose $d = r/2$, $L = 3$, $h_1 = 2d$, $h_2 = 3d$, and $\eta = 1$, we have $\psi_1 = c\xi\big(|\Omega|^{-1}(2n_2 r + 2r^2)\log(\kappa)\big)^{1/4}$ and $\psi_2 = c\xi\big(|\Omega|^{-1}(2n_2 r)\log(\kappa')\big)^{1/4}$, where $\psi_1 > \psi_2$. It means the upper bound given by the classical MF is tighter than that given by the nonlinear DMF. In general, the superiority of the nonlinear DMF over the classical MF increases when $n_2/n_1$ increases, which is consistent with the numerical results shown in Figure 5 of the appendix.

## 3 TWO-MODE NONLINEAR DEEP MATRIX FACTORIZATION

Data matrices in many problems are square or nearly square, which restricts the capability of nonlinear DMF. For instance, the difference between the width and height of an image is often small. The covariance matrices and graph adjacency matrices are square matrices. In recommendation system, the number of users and the number of items may be of the same order of magnitude. A covariance matrix or graph adjacency matrix may be constructed naturally by data with nonlinear latent structures. Similarly, one may imagine that the rating given by a user on an item is an interaction of the user's features and item's features and the features may have some nonlinear low-dimensional latent structures. The nonlinear DMF in (5) exploits only partial nonlinearity of the data.

Formally, we make the following assumption.

**Assumption 2.** *Suppose* $Y = X + E$, *where* $Y \in \mathbb{R}^{n_1 \times n_2}$ *and* $X = U^\top Q V$. *The columns of* $U \in \mathbb{R}^{m_1 \times n_1}$ *and* $V \in \mathbb{R}^{m_2 \times n_2}$ *are generated respectively by*

$$u = f_1(z) \quad and \quad v = f_2(s),$$

*where* $f_1 : \mathbb{R}^{d_1} \to \mathbb{R}^{m_1}$ *and* $f_2 : \mathbb{R}^{d_2} \to \mathbb{R}^{m_2}$ *are nonlinear mappings,* $d_1 < \min(m_1, n_1)$, $d_2 < \min(m_2, n_2)$, *and* $z \in \mathbb{R}^{d_1}$ *and* $s \in \mathbb{R}^{d_2}$ *denote latent variables. The entries of* $Q \in \mathbb{R}^{m_1 \times m_2}$ *and* $E \in \mathbb{R}^{n_1 \times n_2}$ *are drawn from* $\mathcal{N}(0, \sigma_Q^2)$ *and* $\mathcal{N}(0, \sigma_E^2)$ *respectively.*

Suppose we have a matrix $Y$ given by Assumption 2, we can factorize it as

$$Y \approx \hat{U}^\top \hat{Q} \hat{V}, \quad \hat{u}_{i_1} = f_{\theta_1}(\hat{z}_{i_1}), \quad \hat{v}_{i_2} = f_{\theta_2}(\hat{s}_{i_2}), \quad i_1 = 1, \ldots, n_1, \quad i_2 = 1, \ldots, n_2. \tag{8}$$

In (8), $f_{\theta_1} : \mathbb{R}^{d_1} \to \mathbb{R}^{m_1}$ and $f_{\theta_2} : \mathbb{R}^{d_2} \to \mathbb{R}^{m_2}$ are multi-layer neural networks with parameter sets $\theta_1 = \{W_1^{(1)}, \ldots, W_{L_1}^{(1)}\}$ and $\theta_2 = \{W_1^{(2)}, \ldots, W_{L_2}^{(2)}\}$, where $W_i^{(j)}$ denotes the weight matrix in the $i$-th layer of neural network $j$, and $L_j$ denotes the number of layers of neural network $j$, $j = 1, 2$.

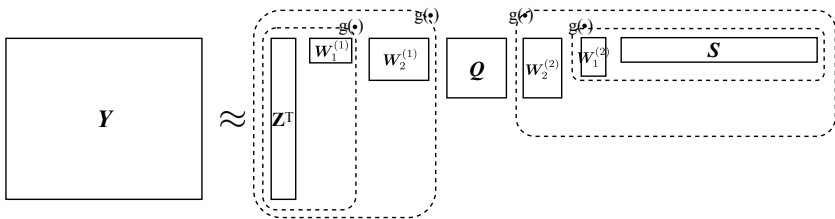

Figure 2: Two-mode nonlinear deep matrix factorization

We call (8) two-mode or two-way nonlinear deep matrix factorization. Figure 2 shows an intuitive example of (8) with depth-three factorizations in both directions.

Suppose we observed a few entries of $\boldsymbol{Y}$ randomly (sampling without replacement). Based on Assumption 2, we propose the following matrix completion method

$$\underset{\boldsymbol{Z},\boldsymbol{Q},\boldsymbol{S},\theta_1,\theta_2}{\text{minimize}} \ \frac{1}{2}\|\mathcal{P}_\Omega(\boldsymbol{Y} - f_{\theta_1}(\boldsymbol{Z})^\top \boldsymbol{Q} f_{\theta_2}(\boldsymbol{S}))\|_F^2 + R(\boldsymbol{Z},\boldsymbol{Q},\boldsymbol{S},\theta_1,\theta_2), \qquad (9)$$

where

$$f_{\theta_1}(\boldsymbol{Z}) = g(\boldsymbol{W}_{L_1}^{(1)}g(\boldsymbol{W}_{L_1-1}^{(1)}\cdots g(\boldsymbol{W}_1^{(1)}\boldsymbol{Z})\cdots)),$$

$$f_{\theta_2}(\boldsymbol{S}) = g(\boldsymbol{W}_{L_2}^{(2)}g(\boldsymbol{W}_{L_2-1}^{(2)}\cdots g(\boldsymbol{W}_1^{(2)}\boldsymbol{S})\cdots)),$$

and $\boldsymbol{Z} \in \mathbb{R}^{d_1 \times n_1}$, $\boldsymbol{S} \in \mathbb{R}^{d_2 \times n_2}$, $\boldsymbol{W}_l^{(j)} \in \mathbb{R}^{h_l^{(j)} \times h_{l-1}^{(j)}}$, $l = 1,\ldots,L_j$, $h_{L_j}^{(j)} = m_j$, $h_0^{(j)} = d_j$, $h_{-1}^{(j)} = n_j$, $j = 1,2$, and $\boldsymbol{Q} \in \mathbb{R}^{m_1 \times m_2}$. We have let all activation functions be the same without loss of generality. In (9), $R(\cdot)$ is a regularization operator to reduce overfitting. For instance, $R(\boldsymbol{Z},\boldsymbol{Q},\boldsymbol{S},\theta_1,\theta_2) = \frac{\lambda_1}{2}(\|\boldsymbol{Z}\|_F^2 + \|\boldsymbol{S}\|_F^2) + \frac{\lambda_2}{2}\left(\|\boldsymbol{Q}\|_F^2 + \sum_{\boldsymbol{W} \in \theta_1 \cup \theta_2}\|\boldsymbol{W}\|_F^2\right)$. Compared to (5), (9) exploits the full nonlinearity of the data. The following theorem provides the excess risk for (9).

**Theorem 2.** *Suppose $\boldsymbol{Z}$, $\boldsymbol{S}$, $\{\boldsymbol{W}_l^{(1)}\}_{l=1}^{L_1}$, $\{\boldsymbol{W}_l^{(2)}\}_{l=1}^{L_2}$, and $\boldsymbol{Q}$ are given by (9). Let $\hat{\boldsymbol{X}} = f_{\theta_1}(\boldsymbol{Z})^\top \boldsymbol{Q} f_{\theta_2}(\boldsymbol{S})$. Suppose $\|\boldsymbol{Q}\|_F\|\boldsymbol{Z}\|_F\|\boldsymbol{S}\|_F \prod_{j=1}^2 \prod_{l=1}^{L_j}\|\boldsymbol{W}_l^{(j)}\|_F \leq \kappa$, $\max(\|\boldsymbol{Y}\|_\infty, \|\hat{\boldsymbol{X}}\|_\infty) \leq \xi$, and the Lipschitz constant of $g$ is $\eta$. Let $N = n_1 n_2$. Then with probability at least $1 - 2N^{-1}$, there exists a numerical constant $c$ such that*

$$\frac{1}{\sqrt{N}}\|\boldsymbol{X} - \hat{\boldsymbol{X}}\|_F \leq \frac{1}{\sqrt{|\Omega|}}\|\mathcal{P}_\Omega(\boldsymbol{Y} - \hat{\boldsymbol{X}})\|_F + \frac{1}{\sqrt{N}}\|\boldsymbol{E}\|_F$$

$$+ c\xi \left(\frac{\left(m_1 m_2 + \sum_{j=1}^2 \sum_{l=0}^{L_j} h_l^{(j)} h_{l-1}^{(j)}\right) \log(\eta^{L_1+L_2}\xi^{-1}\kappa)}{|\Omega|}\right)^{1/4}.$$

In Theorem 2, the last term of the RHS of the inequality can be written as

$$\psi_3 = c\xi\left(|\Omega|^{-1}(n_1 d_1 + n_2 d_2 + m_1 m_2 + \sum_{j=1}^2 \sum_{l=1}^{L_j} h_l^{(j)} h_{l-1}^{(j)})\log(\eta^{L-1}\xi^{-1}\kappa)\right)^{1/4}. \qquad (10)$$

When the $h_{L-1}$ in (6) is much larger than the $d_1$ in (10), we have $\psi_3 < \psi_1$. It means the two-mode nonlinear DMF (9) provides a tighter generalization bound than classical nonlinear DMF (5).

It is worth mentioning that one may formulate DMF in a two-sided form, e.g., $\boldsymbol{X} \approx g(\boldsymbol{W}_L^{(2)}g(\boldsymbol{W}_L^{(1)}\boldsymbol{Z}\boldsymbol{W}_R^{(1)})\boldsymbol{W}_R^{(2)})$ or $\boldsymbol{X} \approx g(\boldsymbol{W}_L^{(1)}g(\boldsymbol{Z}_L \boldsymbol{Z}_R)\boldsymbol{W}_R^{(1)})$ equivalently. However, this model has two shortcomings compared to the one-sided DMF (4) and two-mode DMF (8). First, (4) and (8) are more compact (in terms of the number of parameters) than the two-sided DMF. For example, assume the sizes of $\boldsymbol{X}$, $\boldsymbol{Z}_L$, and $\boldsymbol{Z}_R$ are 500×500, 100×10, and 10×100 respectively, then the number of parameters in the two-sided DMF is larger than 100,000. In the one-sided DMF (4), let $\boldsymbol{X} = g(\boldsymbol{W}_2 g(\boldsymbol{W}_1 \boldsymbol{Z}))$ and let the sizes of $\boldsymbol{W}_1$, $\boldsymbol{W}_2$, and $\boldsymbol{Z}$ be 500×100, 100×10, and 10×500 respectively, then the number of parameters is 56,000, which is much smaller than that of the two-sided DMF. The other shortcoming is that the two-sided DMF is difficult to interpret in real applications. In contrast, (4) and (8) are much easier to interpret and more realistic.

## 4 MULTI-MODE NONLINEAR DEEP TENSOR FACTORIZATION

In this section, we extend the two-mode deep matrix factorization to multi-mode nonlinear deep tensor factorization. First, we make the following assumption.

**Assumption 3.** *Suppose $\boldsymbol{\mathcal{Y}} = \boldsymbol{\mathcal{X}} + \boldsymbol{\mathcal{E}}$, where $\boldsymbol{\mathcal{Y}} \in \mathbb{R}^{n_1 \times n_2 \cdots \times n_k}$ and $\boldsymbol{\mathcal{X}} = \boldsymbol{\mathcal{C}} \times_1 \boldsymbol{U}_1 \times_2 \boldsymbol{U}_2 \cdots \times_k \boldsymbol{U}_k$. For $j = 1, \ldots, k$, the rows of $\boldsymbol{U}_j \in \mathbb{R}^{n_j \times m_j}$ are generated by $\boldsymbol{u}^{(j)} = f_j(\boldsymbol{z}^{(j)})$, where $f_j : \mathbb{R}^{d_j} \to \mathbb{R}^{m_j}$ is a nonlinear mapping, $d_j < m_j$, and $\boldsymbol{z}^{(j)} \in \mathbb{R}^{d_j}$ are latent variables. The entries of core tensor $\boldsymbol{\mathcal{C}} \in \mathbb{R}^{m_1 \times m_2 \cdots \times m_k}$ and noise tensor $\boldsymbol{\mathcal{E}} \in \mathbb{R}^{n_1 \times n_2 \cdots \times n_k}$ are drawn from $\mathcal{N}(0, \sigma_{\boldsymbol{\mathcal{C}}}^2)$ and $\mathcal{N}(0, \sigma_{\boldsymbol{\mathcal{E}}}^2)$ respectively.*

Although $\{f_j\}_{j=1}^k$, $\boldsymbol{\mathcal{C}}$, and $\{\boldsymbol{Z}^{(j)}\}_{j=1}^k$ are unknown, we can take advantage of neural networks and tensor decomposition to model the data, i.e.,

$$\boldsymbol{\mathcal{Y}} \approx \hat{\boldsymbol{\mathcal{C}}} \times_1 \hat{\boldsymbol{U}}_1 \times_2 \hat{\boldsymbol{U}}_2 \cdots \times_k \hat{\boldsymbol{U}}_k, \quad \hat{\boldsymbol{U}}_j = f_{\theta_j}(\hat{\boldsymbol{Z}}^{(j)}), \; j = 1, \ldots, k, \tag{11}$$

where $f_{\theta_j} : \mathbb{R}^{d_j} \to \mathbb{R}^{d_j \times m_j}$ is a neural network with parameter set $\theta_j$. Figure 3 shows an intuitive example of the proposed factorization method for a third-order tensor. Different from Tucker decomposition, we perform nonlinear DMF on the factor matrices and do not constrain the norms of the columns of the factor matrices. This factorization method can take advantage of the nonlinear structures of the tensor fully.

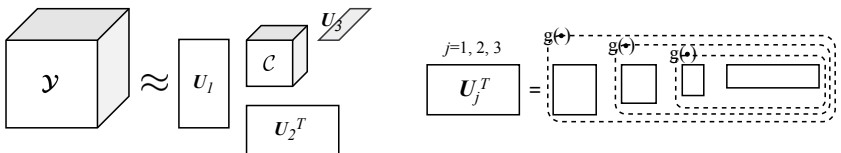

Figure 3: Multi-mode nonlinear deep tensor factorization

To recover $\boldsymbol{\mathcal{X}}$ from a few entries of $\boldsymbol{\mathcal{Y}}$, we propose to solve the following problem

$$\underset{\{\theta_j, \boldsymbol{Z}^{(j)}\}_{j=1}^k, \boldsymbol{\mathcal{C}}}{\text{minimize}} \; \frac{1}{2} \|\mathcal{P}_\Omega(\boldsymbol{\mathcal{Y}} - \boldsymbol{\mathcal{C}} \times_1 f_{\theta_1}(\boldsymbol{Z}^{(1)}) \times_2 f_{\theta_2}(\boldsymbol{Z}^{(2)}) \cdots \times_k f_{\theta_k}(\boldsymbol{Z}^{(k)})\|_F^2 + R(\{\theta_j, \boldsymbol{Z}^{(j)}\}_{j=1}^k, \boldsymbol{\mathcal{C}}), \tag{12}$$

where $f_{\theta_j}(\boldsymbol{Z}^{(j)}) = g(\boldsymbol{W}_{L_j}^{(j)} g(\boldsymbol{W}_{L_j-1}^{(j)} \cdots g(\boldsymbol{W}_1^{(j)} \boldsymbol{Z}^{(j)}) \cdots))$ and $\theta_j = \{\boldsymbol{W}_1^{(j)}, \ldots, \boldsymbol{W}_{L_j}^{(j)}\}$, $j = 1, \ldots, k$. $R(\cdot)$ denotes the regularization on the parameters, e.g.,

$$R(\{\theta_j, \boldsymbol{Z}^{(j)}\}_{j=1}^k, \boldsymbol{\mathcal{C}}) = \frac{\lambda_1}{2} \sum_{j=1}^k \|\boldsymbol{Z}^{(j)}\|_F^2 + \frac{\lambda_2}{2} \Big( \|\boldsymbol{\mathcal{C}}\|_F^2 + \sum_{j=1}^k \sum_{l=1}^{L_j} \|\boldsymbol{W}_l^{(j)}\|_F^2 \Big). \tag{13}$$

The following theorem shows the excess risk bound for (12).

**Theorem 3.** *Suppose $\boldsymbol{\mathcal{C}}$ and $\{\theta_j, \boldsymbol{Z}^{(j)}\}_{j=1}^k$ are given by (12). Let $\hat{\boldsymbol{\mathcal{X}}} = \boldsymbol{\mathcal{C}} \times_1 f_{\theta_1}(\boldsymbol{Z}^{(1)}) \times_2 f_{\theta_2}(\boldsymbol{Z}^{(2)}) \cdots \times_k f_{\theta_k}(\boldsymbol{Z}^{(k)})$. Suppose $\|\boldsymbol{\mathcal{C}}\|_F \leq \beta_C$, $\|\boldsymbol{W}_l^{(j)}\|_F \leq \beta_l^{(j)}$, $l = 1, \ldots, L_j$, $\|\boldsymbol{Z}^{(j)}\|_F \leq \beta_0^{(j)}$, $j = 1, \ldots, k$, $\max(\|\boldsymbol{\mathcal{Y}}\|_\infty, \|\hat{\boldsymbol{\mathcal{X}}}\|_\infty) \leq \xi$, and the Lipschitz constant of $g$ is $\eta$. Let $N = \prod_{j=1}^k n_j$. Then with probability at least $1 - 2N^{-1}$, there exists a numerical constant $c$ such that*

$$\frac{1}{\sqrt{N}} \|\boldsymbol{\mathcal{X}} - \hat{\boldsymbol{\mathcal{X}}}\|_F \leq \frac{1}{\sqrt{|\Omega|}} \|\mathcal{P}_\Omega(\boldsymbol{\mathcal{Y}} - \hat{\boldsymbol{\mathcal{X}}})\|_F + \frac{1}{\sqrt{N}} \|\boldsymbol{\mathcal{E}}\|_F$$

$$+ c\xi \left( \frac{\left( \prod_{j=1}^k m_j + \sum_{j=1}^k \sum_{l=0}^{L_j} h_l^{(j)} h_{l-1}^{(j)} \right) \log \left( \eta^{\sum_{j=1}^k L_j} \xi^{-1} \beta_C \prod_{j=1}^k \prod_{l=0}^{L_j} \beta_l^{(j)} \right)}{|\Omega|} \right)^{1/4}.$$

**Remark 1.** *One can obtain slightly tighter bounds in Theorems 1, 2, and 3 when using spectral norms instead of Frobenius norms. But in (5), (9), and (12), the regularizations only control the Frobenius norms explicitly. In addition, (12) and Theorem 3 are also applicable to CP-like decomposition based tensor completion if $\boldsymbol{\mathcal{C}}$ is super-diagonal.*

We see that Theorem 2 is a special case of Theorem 3 when $k = 2$. Denote $\psi_4$ the last term of the RHS of the inequality in Theorem 3. Since $h_0^{(j)} = d_j$ and $h_{-1}^{(j)} = n_j$, we have

$$\psi_4 = c\xi \left( |\Omega|^{-1} \Big( \prod_{j=1}^{k} m_j + \sum_{j=1}^{k} n_j d_j + \sum_{j=1}^{k} \sum_{l=1}^{L_j} h_l^{(j)} h_{l-1}^{(j)} \Big) \log \Big( \eta^{\sum_{j=1}^{k} L_j} \xi^{-1} \beta_C \prod_{j=1}^{k} \prod_{l=0}^{L_j} \beta_l^{(j)} \Big) \right)^{1/4}.$$
(14)

If $g$ is linear and $L = 0$, (12) degenerates into the Tucker decomposition based LRTC, e.g. (3). Accordingly, $\psi_4$ becomes

$$\psi_5 = c\xi \left( |\Omega|^{-1} \Big( \prod_{j=1}^{k} m_j + \sum_{j=1}^{k} n_j m_j \Big) \log \Big( \xi^{-1} \beta_C \prod_{j=1}^{k} \beta_0^{(j)} \Big) \right)^{1/4}.$$
(15)

Since $m_j \gg d_j$, we have $\psi_5 > \psi_4$. It means the nonlinear method (12) has a tighter generalization error bound than linear tensor completion methods. Therefore, we expect that (12) outperforms linear LRTC methods when the data have nonlinear structures. For convenience, in this paper, we call (9) and (12) multi-mode deep matrix and tensor factorizations ($M^2$DMTF).

**Optimization**  The optimization for (9) and (12) are non-trivial because there are high-order tensors and neural networks with unknown inputs and incomplete outputs. We tried alternating minimization, nonlinear conjugate descent, L-BFGS (Liu & Nocedal, 1989), iRprop+ (Igel & Hüsken, 2000), and Adam (Kingma & Ba, 2014), where all variables are randomly initialized from Gaussian distribution. It is found that iRProp+ outperforms other algorithms especially on real datasets. The details of optimization and the analysis of time and space complexity are presented in Appendix A.

## 5 NUMERICAL RESULTS

### 5.1 SYNTHETIC DATA

We use the following nonlinear model to generate matrices:

$$\boldsymbol{U} = \boldsymbol{W}_3^{(1)} \sigma(\boldsymbol{W}_2^{(1)} \sin(\boldsymbol{W}_1^{(1)} \boldsymbol{Z})), \quad \boldsymbol{V} = \boldsymbol{W}_3^{(2)} \exp(\boldsymbol{W}_2^{(2)} \sigma(\boldsymbol{W}_1^{(2)} \boldsymbol{S})), \quad \boldsymbol{X} = \boldsymbol{U}^\top \boldsymbol{V}, \quad (16)$$

where $\boldsymbol{Z} \in \mathbb{R}^{2 \times 100}$, $\boldsymbol{W}_1^{(1)} \in \mathbb{R}^{10 \times 2}$, $\boldsymbol{W}_2^{(1)} \in \mathbb{R}^{20 \times 10}$, $\boldsymbol{W}_3^{(1)} \in \mathbb{R}^{30 \times 20}$, $\boldsymbol{S} \in \mathbb{R}^{2 \times 100}$, $\boldsymbol{W}_1^{(2)} \in \mathbb{R}^{10 \times 2}$, $\boldsymbol{W}_2^{(2)} \in \mathbb{R}^{20 \times 10}$, and $\boldsymbol{W}_3^{(2)} \in \mathbb{R}^{30 \times 20}$ are drawn from $\mathcal{U}(-1, 1)$. $\sigma$ denotes the sigmoid fucntion. The exact rank of $\boldsymbol{X}$ is 20 but the smallest 10 nonzero singular values are close to zero. Such a synthetic matrix is realistic because the nonlinearity in real data may not be very strong and the matrices are often low-rank. Similarly, the following nonlinear model is used to generate third-order tensors:

$$\boldsymbol{U}_1 = \sigma(\boldsymbol{W}_2^{(1)} \sin(\boldsymbol{W}_1^{(1)} \boldsymbol{Z}^{(1)})), \quad \boldsymbol{U}_2 = \exp(\boldsymbol{W}_2^{(2)} \sigma(\boldsymbol{W}_1^{(2)} \boldsymbol{Z}^{(2)})),$$
$$\boldsymbol{U}_3 = \cos(\boldsymbol{W}_2^{(3)} \exp(\boldsymbol{W}_1^{(3)} \boldsymbol{Z}^{(3)})), \quad \mathcal{X} = \mathcal{C} \times_1 \boldsymbol{U}_1 \times_2 \boldsymbol{U}_2 \times_3 \boldsymbol{U}_3,$$
(17)

where $\boldsymbol{Z}^{(1)} \in \mathbb{R}^{2 \times 50}$, $\boldsymbol{W}_1^{(1)} \in \mathbb{R}^{10 \times 2}$, $\boldsymbol{W}_2^{(1)} \in \mathbb{R}^{20 \times 10}$, $\boldsymbol{Z}^{(2)} \in \mathbb{R}^{2 \times 50}$, $\boldsymbol{W}_1^{(2)} \in \mathbb{R}^{10 \times 2}$, $\boldsymbol{W}_2^{(2)} \in \mathbb{R}^{20 \times 10}$, $\boldsymbol{Z}^{(3)} \in \mathbb{R}^{2 \times 50}$, $\boldsymbol{W}_1^{(3)} \in \mathbb{R}^{10 \times 2}$, and $\boldsymbol{W}_2^{(3)} \in \mathbb{R}^{20 \times 10}$ are drawn from $\mathcal{U}(-1, 1)$ and $\mathcal{C} \in \mathbb{R}^{20 \times 20 \times 20}$ are drawn from $\mathcal{N}(0, 1)$. The exact $n$-rank of $\mathcal{X}$ is $(20, 20, 20)$ but it can be well approximated by a tensor with $n$-rank $(10, 10, 10)$ because here the nonlinearity is not very strong.

First, we compare the proposed method with linear matrix factorization (MF (1)), nuclear norm minimization (NNM) (Candès & Recht, 2009), TNNR (Hu et al., 2013), HM-IRLS (Kümmerle & Sigl, 2018), FGSR (Fan et al., 2019), (stacked) Autoencoders (SAE) (Sedhain et al., 2015), and DMF (Fan & Cheng, 2018). We use the $tanh$ activation function in SAE, DMF, and $M^2$DMTF. We determine the hyper parameters of all methods via cross-validation to ensure they perform as well as possible. Details about the optimizations and parameter settings are in Appendix B. We use the relative recovery error (Fan et al., 2019) $\|\mathcal{P}_{\bar{\Omega}}(\hat{\boldsymbol{X}} - \boldsymbol{X})\|_F / \|\mathcal{P}_{\bar{\Omega}}(\boldsymbol{X})\|_F$ to evaluate the performance of matrix completion, where $\bar{\Omega}$ consists of the locations of the missing entries. Figure 4(a) shows the average performance of 20 repeated trials when the missing rate increases from 0.8 to 0.99. MF

and NNM have much higher recovery error than FGSR. This result is consistent with the results of (Fan et al., 2019; Arora et al., 2019). Although SAE is a nonlinear method, it still has high recovery error when the missing rate is less than 0.9. The reason is that filling the missing entries of the input of the encoder with zero introduces relatively large bias, which may not be eliminated effectively by the hidden layers. DMF and FGSR have quite similar performance, which is consistent with the analysis at the end of Section 2. Our M$^2$DMTF outperformed other methods significantly especially when the missing rate is high. The reason is that M$^2$DMTF is able to exploit the full nonlinearity of the data and has tighter generalization bound than other methods.

Figure 4(b) shows the performance (average of 20 trials) of FaLRTC (Liu et al., 2012), TenALS (Jain & Oh, 2014), TMac (Xu et al., 2013), KBR-TC (Xie et al., 2018), TRLRF (Yuan et al., 2019), CoSTCo (Liu et al., 2019), OITNN-O (Wang et al., 2020), and M$^2$DMTF. Since the rank of the tensor is quite low, TMac, KBR-TC, and M$^2$DMTF have high recovery accuracy when the missing rate is less than 0.96. Note that TMac and KBR-TC slightly outperformed M$^2$DMTF when the missing rate is less than 0.96. The reason is that the optimization of M$^2$DMTF is more difficult. In Figure 4(b), when the missing rate is higher than 0.97, the proposed M$^2$DMTF outperformed all methods largely. The optimization curves of M$^2$DMTF are shown in Figure 7 of Appendix B.

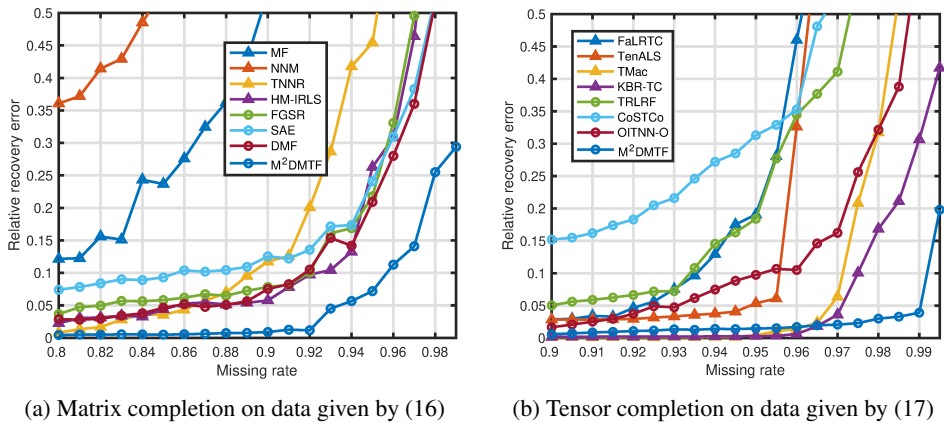

(a) Matrix completion on data given by (16)    (b) Tensor completion on data given by (17)

Figure 4: Performance evaluation of matrix completion and tensor completion on synthetic data.

## 5.2 REAL MATRICES

We consider two benchmark datasets: MovieLens-100k and MovieLens-1M. They consist of the ratings (1 to 5) for movies given by users. The sizes (movies×users) of the rating matrices in the datasets are $1682 \times 943$ and $3706 \times 6040$ respectively. For each matrix (highly incomplete), we randomly split the known entries to a training set and a test set. The relative recovery errors (average of 10 repeated trials) on the two datasets are reported in Tables 1 and 2 respectively. FGSR, SAE, and M$^2$DMTF outperformed MF and DMF in all cases. When the training-test ratio is 9/1 or 7/3, SAE has less recovery error than M$^2$DMTF. This is quite different from the results on the synthetic data in Section 5.1. Currently, it is unknown why SAE has such a good performance on the MovieLens data. One possible reason is that the MovieLens data are quite noisy and SAE can successfully reduce the influence of the noise when the missing rate is relatively low. When the training-test ratio is smaller, i.e., 5/5, 3/7, or 1/9, M$^2$DMTF outperformed SAE.

Table 1: Relative recovery error of matrix completion on MovieLens-100k

| Train/Test | MF | FGSR | SAE | DMF | M$^2$DMTF |
|---|---|---|---|---|---|
| 9/1 | 0.2584±0.0019 | 0.2483±0.0020 | **0.2412**±0.0017 | 0.2518±0.0020 | 0.2445±0.0020 |
| 7/3 | 0.2666±0.0011 | 0.2514±0.0009 | **0.2456**±0.0012 | 0.2551±0.0011 | 0.2469±0.0014 |
| 5/5 | 0.2795±0.0010 | 0.2575±0.0011 | 0.2527±0.0008 | 0.2603±0.0014 | **0.2519**±0.0011 |
| 3/7 | 0.3039±0.0009 | 0.2679±0.0009 | 0.2656±0.0004 | 0.2738±0.0007 | **0.2580**±0.0007 |
| 1/9 | 0.3940±0.0025 | 0.3233±0.0026 | 0.3028±0.0026 | 0.3157±0.0018 | **0.2731**±0.0008 |

Table 2: Relative recovery error of matrix completion on on MovieLens-1M

| Train/Test | MF | FGSR | SAE | DMF | M$^2$DMTF |
|---|---|---|---|---|---|
| 9/1 | 0.2278±0.0003 | 0.2248±0.0002 | **0.2239**±0.0003 | 0.2266±0.0002 | 0.2261±0.0001 |
| 7/3 | 0.2315±0.0003 | **0.2271**±0.0001 | 0.2278±0.0002 | 0.2291±0.0002 | 0.2276±0.0001 |
| 5/5 | 0.2381±0.0003 | 0.2319±0.0002 | 0.2346±0.0003 | 0.2333±0.0002 | **0.2314**±0.0001 |
| 3/7 | 0.2520±0.0003 | 0.2379±0.0003 | 0.2458±0.0002 | 0.2419±0.0003 | **0.2364**±0.0002 |
| 1/9 | 0.3226±0.0008 | 0.2697±0.0017 | 0.2796±0.0006 | 0.2573±0.0006 | **0.2510**±0.0002 |

## 5.3 REAL TENSORS

We compare the proposed method M$^2$DMTF with the baselines on the following datasets: Amino acid fluorescence (Bro, 1997) ($5 \times 201 \times 61$), Flow injection (Nørgaard & Ridder, 1994) ($12 \times 100 \times 89$), and SW-NIR kinetic data (Bijlsma & Smilde, 2000) ($301 \times 241 \times 8$). More details about the experimental settings are in Appendix B.2. The relative recovery error (average of 10 trials) are reported in Table 3. The results of FaLRTC and CoSTCo are not reported because they have much higher recovery error than other methods in most cases and we want to fit the table to the width of the page. On Amino, our M$^2$DMTF outperformed other methods significantly in all cases. On Flow, when the missing rate is 0.95 or 0.97, M$^2$DMTF has much lower recovery error than other methods. On SW-NIR, M$^2$DMTF has low recovery error consistently while most baselines failed totally.

Table 3: Relative recovery error on three real tensor datasets (MR denotes missing rate)

| Data | MR | TenALS | TMac | KBR-TC | TRLRF | OITNN-O | M$^2$DMTF |
|---|---|---|---|---|---|---|---|
| Amino | 0.85 | 0.0252±0.0019 | 0.0225±0.0021 | 0.0318±0.0011 | 0.0486±0.0180 | 0.0348±0.0016 | **0.0189**±0.0007 |
| | 0.90 | 0.1317±0.0208 | 0.0451±0.0063 | 0.0358±0.0022 | 0.1004±0.0284 | 0.0684±0.0096 | **0.0213**±0.0010 |
| | 0.95 | 0.4727±0.0213 | 0.3520±0.1632 | 0.1370±0.0131 | 0.3941±0.1029 | 0.2803±0.0246 | **0.0354**±0.0009 |
| | 0.97 | > 0.95 | 0.5209±0.0926 | 0.3192±0.0453 | 0.5810±0.1136 | 0.5651±0.0309 | **0.1387**±0.0346 |
| Flow | 0.85 | 0.0344±0.0004 | 0.0128±0.0001 | **0.0028**±0.0001 | 0.0471±0.0079 | 0.0253±0.0016 | 0.0206±0.0045 |
| | 0.90 | 0.0359±0.0044 | 0.0132±0.0002 | **0.0041**±0.0002 | 0.0619±0.0117 | 0.0451±0.0046 | 0.0198±0.0054 |
| | 0.95 | 0.3495±0.1838 | 0.0716±0.0329 | 0.0695±0.0322 | 0.3817±0.0964 | 0.1369±0.0088 | **0.0396**±0.0127 |
| | 0.97 | 0.4141±0.1678 | 0.3488±0.0405 | 0.3627±0.0509 | 0.5985±0.1081 | 0.2617±0.0176 | **0.0503**±0.0185 |
| SW-NIR | 0.85 | 0.2894±0.0190 | 0.5409±0.0160 | 0.6546±0.0174 | 0.2425±0.0281 | 0.1464±0.0309 | **0.0593**±0.0114 |
| | 0.90 | 0.3427±0.0144 | 0.8760±0.1298 | 0.6845±0.0327 | 0.3607±0.0360 | 0.4683±0.0617 | **0.0601**±0.0078 |
| | 0.95 | 0.4175±0.0246 | > 0.95 | 0.8334±0.0291 | 0.9146±0.0446 | 0.8881±0.0231 | **0.0758**±0.0023 |
| | 0.97 | > 0.95 | 0.8823 | > 0.95 | > 0.95 | > 0.95 | **0.0862**±0.0057 |

Comparative studies were also conducted on three larger datasets. The results are reported in Appendix B.2.4. The experiments of tensor completion based recommendation system are in Section B.2.5. The visualization of the learned tensor and factors of M$^2$DMTF are in Appendix B.2.6.

## 6 CONCLUSION

This paper has proposed a framework of multi-mode deep matrix and tensor factorizations[1]. The proposed factorization methods can explore and exploit the full nonlinearity of the data effectively. We applied the factorization methods to matrix and tensor completion and provided the generalization error bounds, which verified the superiority of the proposed methods over conventional linear and nonlinear matrix factorization methods and linear tensor completion methods. The proposed framework reduced the gap between tensor decomposition and deep learning and showed that their combinations can provide better performance in tensor recovery. On some datasets, when the missing rates were low, our methods did not achieve the best performance. The reason is that the nonlinearity in the data is not very strong and the problem is not difficult for the baselines methods of which the optimizations are easier than our methods. Future work may focus on improving the optimization efficiency of M$^2$DMTF.

---

[1]Codes link: https://github.com/jicongfan/Multi-Mode-Deep-Matrix-and-Tensor-Factorization

ACKNOWLEDGEMENTS

The work was supported by the research funding T00120210002 of Shenzhen Research Institute of Big Data and the Youth program 62106211 of National Natural Science Foundation of China.

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

## A    OPTIMIZATION DETAILS AND COMPUTATIONAL COMPLEXITY

In our M$^2$DMTF, the matrix model is a special case of the tensor model. Therefore, we only show the optimization detail and the computational complexity of the tensor case. The gradient-based optimization is shown in Algorithm 1. We suggest using iRprop+ as the optimizer in the algorithm because we found empirically that iRprop+ is more stable and efficient than other optimizers such as Adam in solving our M$^2$DMTF.

---

**Algorithm 1** Gradient-based optimization for M$^2$DMTF (12)

---

**Input:** $\mathcal{Y} \in \mathbb{R}^{n_1 \times \cdots \times n_k}$, $\Omega$, $\lambda_1$, $\lambda_2$, $g$, network structures $\{m_j, L_j, \{h_l^{(j)}\}_{l=1}^{L_j}\}_{j=1}^{k}$, $T$.

1: Initialization: $\mathcal{C}_0$, $\{Z_0^{(j)}, \{W_{l0}^{(j)}\}_{l=1}^{L_j} \triangleq \theta_0^{(j)}\}_{j=1}^{k}$, $\{U_0^{(j)} = f_{\theta_0^{(j)}}(Z_0^{(j)})\}_{j=1}^{k}$.

2: **for** $t = 1, 2, \ldots, T$ **do**

3:     **for** $j = 1, 2, \ldots, k$ **do**

        $\{\nabla_{\theta^{(j)}}, \nabla_{Z^{(j)}}\} \xleftarrow{\text{backprop.}} \mathcal{L}_j$, where $\mathcal{L}_j = \dfrac{1}{2}\left\|\mathcal{P}_{\Omega_{(j)}}\left(\mathcal{Y}_{(j)} - U_t^{(j)}[\mathcal{C}_t]_{(j)} V_t^\top\right)\right\|_F^2 +$

    $R(\{\theta_t^{(j)}, Z^{(j)}\}_{j=1}^k, \mathcal{C}_t)$ and $V_t = U_t^{(j-1)} \otimes \cdots U_1^{(1)} \otimes U_t^{(k)} \cdots \otimes U_t^{(j+1)}$.

4:     **end for**

5:     $\nabla_{\mathcal{C}} \longleftarrow \partial \mathcal{L}_k / \partial \mathcal{C}$

6:     $\{\{\theta_t^{(j)}, Z_t^{(j)}\}_{j=1}^k, \mathcal{C}_t\} \longleftarrow$ gradient-based optimizer $\left(\{\nabla_{\theta^{(j)}}, \nabla_{Z^{(j)}}\}_{j=1}^k, \nabla_{\mathcal{C}}\right)$

7: **end for**

**Output:** $\hat{\mathcal{Y}} = \mathcal{C}_T \times_1 U_T^{(1)} \times_2 U_T^{(2)} \cdots \times_k U_T^{(k)}$.

---

Without loss of generality, we here assume that in (12), $n_1 = \cdots n_k = n$, $m_1 = \cdots m_k = m$, $L_1 = \cdots = L_k = L$, and $h_l^{(1)} = \cdots = h_l^{(k)} = h_l$, $l = 0, 1, \ldots, L$. The time cost in each iteration of the gradient-based optimization is

$$\mathcal{O}\left( k \min\{nm^k + n^k m^{k-1}, n^k m + n^{k-1} m^k\} + kn \sum_{l=1}^{L} h_{l-1} h_l \right),$$

in which the first term is from the Tucker-like decomposition while the second term is from the forward and backward propagations of the neural networks. Since the second term is much smaller than the first one and $m < n$, the time complexity per iteration is about $\mathcal{O}\left( kn^k m + kn^{k-1} m^k \right)$. As we need to store the observed elements of $\mathcal{Y}$, core tensor $\mathcal{C}$, latent factors $\boldsymbol{Z}^{(j)}$, network parameter, their gradients, and the values of hidden units, the space complexity is $\mathcal{O}\big(|\Omega| + m^k + k \sum_{l=1}^{L} h_{l-1} h_l + kn \sum_{l=1}^{L} h_l\big)$, which is further simplified to $\mathcal{O}\big(|\Omega| + m^k + kn \sum_{l=1}^{L} h_l\big)$ because $n$ is larger than all $h_l$. Now we see that the time (per iteration in the optimization) and space complexity of our M$^2$DMTF are slightly higher than the Tucker decomposition based tensor completion methods. However, M$^2$DMTF requires much more iterations than the baseline methods because it is composed of tensor decomposition and neural networks.

## B    MORE ABOUT THE EXPERIMENTS

### B.1    MATRIX COMPLETION

In the proposed method (9), the regularization is

$$R(\boldsymbol{Z}, \boldsymbol{Q}, \boldsymbol{S}, \theta_1, \theta_2) = \frac{\lambda_1}{2}(\|\boldsymbol{Z}\|_F^2 + \|\boldsymbol{S}\|_F^2) + \frac{\lambda_2}{2}\left( \|\boldsymbol{Q}\|_F^2 + \sum_{j=1}^{2} \sum_{l=1}^{L_j} \|\boldsymbol{W}_l^{(j)}\|_F^2 \right).$$

We let $\lambda_2 = \lambda_2' \sqrt{n_1 n_2}$ and determine $\lambda_1$ and $\lambda_2'$ instead. In the optimization, $\boldsymbol{S}$, $\boldsymbol{Z}$, and $\{\boldsymbol{W}_l^{(j)}\}$ are initialized randomly, $\boldsymbol{Q}$ is initialized to an identity matrix.

#### B.1.1    PARAMETER SETTING ON THE SYNTHETIC DATA

- In MF (problem (1) in the main paper), the factorization dimension $d$ is 5 because it outperforms other choices. The $\lambda$ is chosen from $\{0.01, 0.1, 1\}$ and the optimizer is iRprop+. The maximum iteration is 2000.
- In FGSR (Fan et al., 2019), we use the variational form of Schatten-$1/2$ norm, set $d = 20$, and choose $\lambda$ from $\{0.01, 0.1\}$. The maximum iteration is 2000.
- In SAE (Sedhain et al., 2015), the network structure is $[100, 20, 10, 5, 10, 20, 100]$. The weight decay parameter is chosen from $\{0.1, 0.5\}$.
- In DMF (Fan & Cheng, 2018), the network structure is $[3, 10, 20, 100]$. The weight decay parameters are chosen from $\{0.01, 0.1\}$.
- In M$^2$DMTF, $L = 2$, $d_1 = d_2 = 3$, $h_1^{(1)} = h_1^{(2)} = 10$, $m_1 = m_2 = 20$, and $\lambda_1 = \lambda_2' = 1$.

In SAE, DMF, and M$^2$DMTF, the activation function is the hyperbolic tangent function. The optimizer is iRprop+ and the maximum iteration is 3000.

#### B.1.2    MORE RESULTS ON THE SYNTHETIC DATA

Figure 5 compares DMF with linear MF (FGSR) in the cases of different $n_2/n_1$. We see that when $n_2/n_1$ is too small, FGSR outperforms DMF. When $n_2/n_1$ increases, the superiority of DMF over FGSR becomes more significant. These empirical results are consistent with our theoretical analysis in Section 2 of the main paper.

Figure 6 shows the performance of M$^2$DMTF with different number of hidden layers and different activation function. it can be found that tanh and sigmoid functions lead to lower recovery error than ReLU in all cases. The main reason is that the data were generated by smooth functions that can be

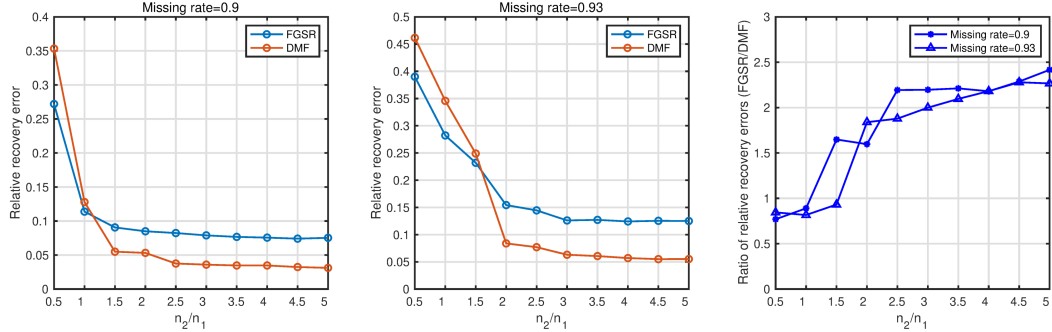

Figure 5: The influence of $n_2/n_1$ on FGSR and DMF

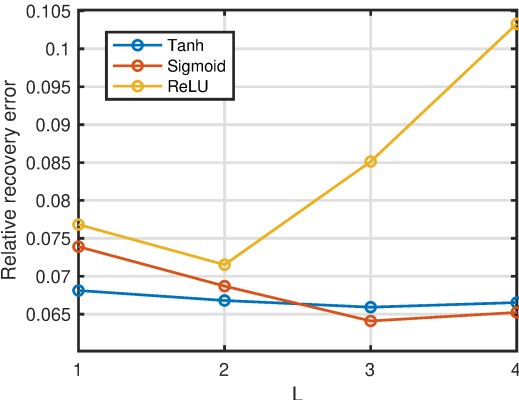

Figure 6: The performance of M$^2$DMTF with different number of hidden layers and different activation function. The sizes of added hidden layers are 10.

well approximated by tanh and sigmoid functions. When the tanh function or sigmoid function is used, increasing the depth of the neural networks can reduce the recovery error. But when $L$ is too large, i.e. $L = 4$, the recovery error increased slightly, which might be caused by overfitting.

Figure 7 shows the values of the objective functions and relative recovery errors against the iterations in matrix completion and tensor completion when the missing rate is $0.95$. The recovery error decreases slightly when the number of iterations is very large. In general, the optimization efficiency is not very high because M$^2$DMTF is composed of multiple neural networks with unknown inputs and incomplete outputs.

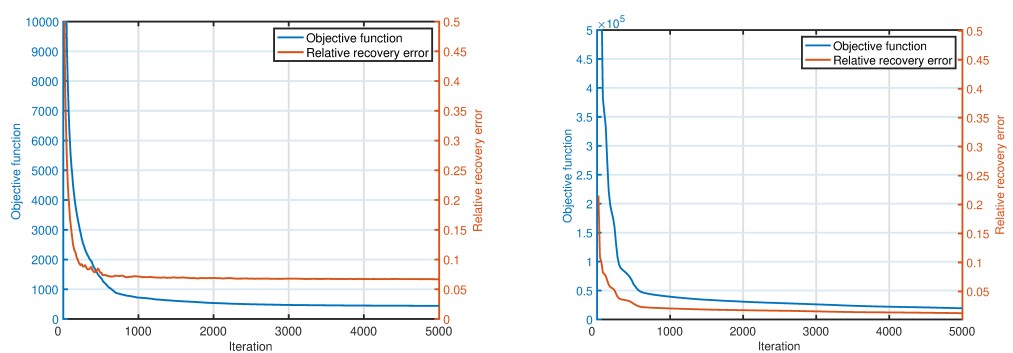

(a) Iterative performance of M$^2$DMTF in matrix completion on data given by (16)

(b) Iterative performance of M$^2$DMTF in tensor completion on data given by (17)

Figure 7: Iterative performance of M$^2$DMTF (iRprop+) on synthetic data with missing rate 0.95.

We replace the sin, cos, exp, and $\sigma$ in (16) and (17) with $tanh$ such that the nonlinearity in the data generating models are consistent with the activation functions used in our matrix and tensor completion method M²DMTF. The average results of 10 trials are reported in Figure 8. We see that M²DMTF outperformed other methods significantly when the missing rate is high enough.

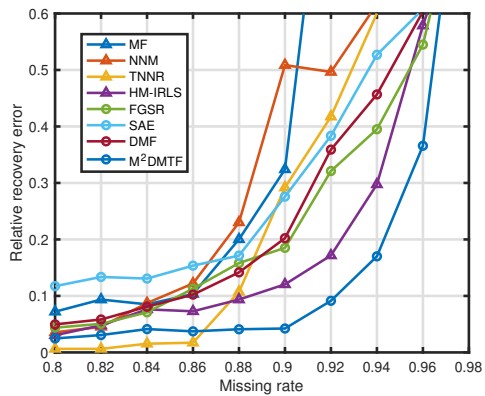 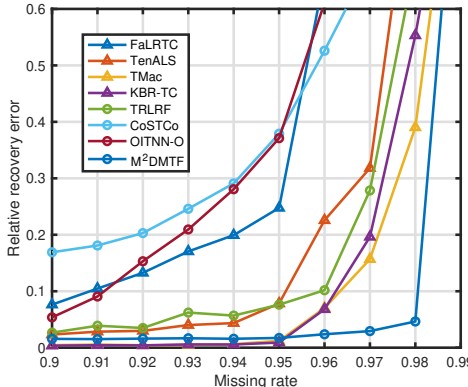

(a) Matrix completion on data given by modified (16)   (b) Tensor completion on data given by modified (17)

Figure 8: Performance evaluation of matrix completion and tensor completion on synthetic data generated by (16) and (17) in which the nonlinear functions replaced by $tanh$.

### B.1.3 PARAMETER SETTING ON THE MOVIELENS DATASETS

- In MF (problem (1) in the main paper), we set $d = 10$ and $\lambda = 10$. In stead of iRprop+, we use alternating minimization because it outperformed iRprop+ in this case. The maximum iteration is 2000.

- In FGSR Fan et al. (2019), we use the variational form of Schatten-$1/2$ norm, set $d = 100$, and set $\lambda = 0.015$ or $0.007$ (for MovieLens-100k or MovieLens-1M). The maximum iteration is 2000.

- In SAE Sedhain et al. (2015), we use a single hidden layer of size 500. The weight decay parameter is $0.5$.

- In DMF, Fan & Cheng (2018), the network structure is $[10 \text{ or } 20, 50, 100, 1682]$. The weight decay parameters are chosen from $\{0.05, 0.1\}$.

- In M²DMTF, , we set $L = 2$, $d = 10$ or $20$, $h_1^{(1)} = h_1^{(2)} = 50$, $m_1 = m_2 = 100$, and $\lambda_1 = \lambda_2' = 1$.

### B.1.4 MORE RESULTS ON THE MOVIELENS DATASETS

The RMSE (average of 10 trials) on the MovieLens-100k are reported in Table 4. The proposed M²DMTF outperformed MF, FGSR and DMF in all cases and outperformed SAE when the missing rate was high. Note that RMSE is a monotonic transformation of the relative recovery error used in the main paper.

Table 4: RMSE on MovieLens-100k

| Train-Test | MF | FGSR | SAE | DMF | M²DMTF |
|---|---|---|---|---|---|
| 9-1 | 0.9565±0.0071 | 0.9194±0.0073 | **0.8928**±0.0063 | 0.9320±0.0070 | 0.9048±0.0071 |
| 7-3 | 0.9880±0.0037 | 0.9318±0.0028 | **0.9105**±0.0036 | 0.9454±0.0038 | 0.9159±0.0045 |
| 5-5 | 1.0356±0.0038 | 0.9541±0.0038 | 0.9364±0.0022 | 0.9692±0.0048 | **0.9341**±0.0040 |
| 3-7 | 1.1259±0.0032 | 0.9924±0.0037 | 0.9840±0.0016 | 1.0155±0.0025 | **0.9563**±0.0025 |
| 1-9 | 1.4597±0.0093 | 1.1978±0.0094 | 1.1217±0.0097 | 1.1697±0.0064 | **1.0119**±0.0025 |

## B.2 TENSOR COMPLETION

In the proposed method (12), the regularization is

$$R(\{\theta_j, \boldsymbol{Z}^{(j)}\}_{j=1}^k, \boldsymbol{\mathcal{C}}) = \frac{\lambda_1}{2} \sum_{j=1}^k \|\boldsymbol{Z}^{(j)}\|_F^2 + \frac{\lambda_2}{2} \Big( \|\boldsymbol{\mathcal{C}}\|_F^2 + \sum_{j=1}^k \sum_{l=1}^{L_j} \|\boldsymbol{W}_l^{(j)}\|_F^2 \Big). \tag{18}$$

We let $\lambda_2 = \lambda_2' \Big( \prod_{j=1}^k n_j \Big)^{1/k}$ and determine $\lambda_1$ and $\lambda_2'$ instead. In the optimization, $\boldsymbol{U}_j$ is initialized by the singular value decomposition of the $j$-mode matricization of $\boldsymbol{\mathcal{Y}}$, $\boldsymbol{\mathcal{C}}$ is initialized to zero, and $\{\boldsymbol{Z}^{(j)}\}$, $\{\boldsymbol{W}_l^{(j)}\}$ are initialized randomly.

### B.2.1 PARAMETER SETTING ON THE SYNTHETIC DATA

- In FaLRTC (Liu et al., 2012), we set $\alpha_1 = \alpha_2 = \alpha_3 = \frac{1}{3}$ choose $\mu$ from $\{0.5, 1, 2, 5, 10\}$.
- In TenALS (Jain & Oh, 2014), the initial rank is 5 or 10 and the max iteration is 200.
- In TMac (Xu et al., 2013), the rank is initialized to $[10, 10, 10]$ or $[5, 5, 5]$ and adjusted adaptively. The maximum iteration is 500.
- In KBR-TC (Xie et al., 2018), $\rho = 1.01$ and $\lambda = 0.01$. The maximum iteration is 300.
- TRLRF (Yuan et al., 2019), the rank is set to $[5, 5, 5]$ or $[10, 10, 10]$. We choose $\lambda$ from $\{0.01, 0.1, 1, 10, 50\}$.
- In CoSTCo (Liu et al., 2019), the learning rate is chosen from $\{0.01, 0.001, 0.001\}$, the rank is chosen from $\{2, 3, 5, 10, 20\}$, the epoch number is 50 or 200, and the batch size is $256 \times 2$ or $256 \times 5$.
- In OITNN-O (Wang et al., 2020), we choose $\alpha$ from $\{0.0001, 0.001, 0.01, 0.1\}$. The max iteration is 500.
- In M$^2$DMTF, $L = 2$, $d_1 = d_2 = d_3 = 3$, $h_1^{(1)} = h_1^{(2)} = h_1^{(3)} = 10$, $m_1 = m_2 = m_3 = 20$, and $\lambda_1 = \lambda_2' = 1$. The optimizer is iRprop+ and the maximum iteration is 3000.

### B.2.2 TIME COST COMPARISON ON THE SYNTHETIC DATA

We report the average time costs (second) and the recovery errors of all methods in Table 5, which also includes the performance of M$^2$DMTF with different iteration numbers. We see that our method with 3000 iterations is slower than other methods. However, when we use less iterations (e.g. 1000 or 200), the time cost of our method is comparable with or even lower than the baselines and its recovery error is still much lower than the baselines.

Table 5: Average time cost (second) on the synthetic tensor data (CoSTCo was performed in Python while all other methods were performed in MATLAB.)

|  | FaLRTC | TenALS | TMac | KBR-TC | TRLRF | CoSTCo | OITNN-O |
|---|---|---|---|---|---|---|---|
| recovery error | 0.944 | 0.973 | 0.323 | 0.169 | 0.621 | 0.617 | 0.328 |
| time cost | 3.5s | 2.1s | 1.4s | 4.7s | 4.3s | 7.2s | 6.8s |

|  | M$^2$DMTF | | | | | | |
|---|---|---|---|---|---|---|---|
| iteration | 100 | 200 | 300 | 500 | 1000 | 2000 | 3000 |
| recovery error | 0.098 | 0.068 | 0.057 | 0.043 | 0.037 | 0.033 | 0.031 |
| time cost | 1.1s | 2.0s | 3.1s | 4.9s | 9.9s | 20.6s | 32.3s |

### B.2.3 EXPERIMENTAL SETTINGS ON REAL TENSORS

We normalize each tensor by $\boldsymbol{\mathcal{X}} \leftarrow \boldsymbol{\mathcal{X}} / \|\boldsymbol{\mathcal{X}}\|_\infty$. In M$^2$DMTF, we set $L = 1$ and $\lambda_1 = \lambda_2' = 0.001$. For $i = 1, 2, 3$, we set $m_i \approx \frac{n_i}{5}$, $d_i \approx \frac{n_i}{10}$ if $n_i > 10$ and set $m_i \approx 0.8 n_i$, $d_i \approx 0.7 n_i$ if $n_i \leq 10$. The parameters of other methods are carefully determined to provide their best performance.

### B.2.4 MORE RESULTS ON TENSOR DATASETS

In addition to the three datasets considered in the main paper, we test the proposed method on three more datasets: Porphyrin (Wold et al., 2006) $170 \times 274 \times 35$, Sugar (Bro, 1999) $256 \times 571 \times 7$, and Bonnie (Acar et al., 2008) $89 \times 97 \times 549$. To reduce the computational costs of all methods on Bonnie, we only consider a subset of size $89 \times 97 \times 100$. The results are reported in Table 6. We see that $M^2$DMTF outperforms other methods when the missing rate is high enough.

Table 6: Relative recovery error on more tensor datasets (MR denotes missing rate)

| Data | MR | TenALS | TMac | KBR-TC | TRLRF | OITNN-O | $M^2$DMTF |
|---|---|---|---|---|---|---|---|
| Porphyrin | 0.85 | $0.0931_{\pm 0.0013}$ | $\mathbf{0.0105}_{\pm 0.0021}$ | $0.0304_{\pm 0.0011}$ | $0.0319_{\pm 0.0180}$ | $0.0118_{\pm 0.0016}$ | $0.0184_{\pm 0.0007}$ |
| | 0.90 | $0.0968_{\pm 0.0383}$ | $0.0281_{\pm 0.0023}$ | $0.0330_{\pm 0.0017}$ | $0.0451_{\pm 0.0053}$ | $0.0225_{\pm 0.0023}$ | $\mathbf{0.0211}_{\pm 0.0018}$ |
| | 0.95 | $0.1091_{\pm 0.0133}$ | $0.0493_{\pm 0.0169}$ | $0.0458_{\pm 0.0030}$ | $0.0701_{\pm 0.0035}$ | $0.0522_{\pm 0.0036}$ | $\mathbf{0.0376}_{\pm 0.0005}$ |
| | 0.97 | $0.1199_{\pm 0.0108}$ | $0.1128_{\pm 0.0138}$ | $0.0569_{\pm 0.0154}$ | $0.1606_{\pm 0.0178}$ | $0.1066_{\pm 0.0052}$ | $\mathbf{0.0450}_{\pm 0.0009}$ |
| Sugar | 0.85 | $0.1438_{\pm 0.0549}$ | $0.0456_{\pm 0.0041}$ | $\mathbf{0.0406}_{\pm 0.0027}$ | $0.1075_{\pm 0.0045}$ | $0.0719_{\pm 0.0046}$ | $0.0504_{\pm 0.0114}$ |
| | 0.90 | $0.1584_{\pm 0.0019}$ | $0.0639_{\pm 0.0034}$ | $\mathbf{0.0563}_{\pm 0.0039}$ | $0.1214_{\pm 0.0076}$ | $0.0915_{\pm 0.0059}$ | $0.0568_{\pm 0.0038}$ |
| | 0.95 | $0.1749_{\pm 0.0208}$ | $0.1173_{\pm 0.0265}$ | $0.1090_{\pm 0.0048}$ | $0.2083_{\pm 0.0196}$ | $0.1573_{\pm 0.0045}$ | $\mathbf{0.0921}_{\pm 0.0043}$ |
| | 0.97 | $> 0.95$ | $0.3129_{\pm 0.0801}$ | $0.2791_{\pm 0.0306}$ | $0.6515_{\pm 0.0364}$ | $0.2443_{\pm 0.0218}$ | $\mathbf{0.1689}_{\pm 0.0316}$ |
| Bonnie | 0.90 | $0.0538_{\pm 0.0032}$ | $0.0084_{\pm 0.0005}$ | $\mathbf{0.0049}_{\pm 0.0001}$ | $0.0407_{\pm 0.0004}$ | $0.0128_{\pm 0.0007}$ | $0.0110_{\pm 0.0002}$ |
| | 0.95 | $0.0598_{\pm 0.0126}$ | $0.0209_{\pm 0.0016}$ | $0.0204_{\pm 0.0022}$ | $0.0633_{\pm 0.0045}$ | $0.0385_{\pm 0.0014}$ | $\mathbf{0.0167}_{\pm 0.0013}$ |
| | 0.97 | $0.3561_{\pm 0.0649}$ | $0.1451_{\pm 0.0386}$ | $0.1005_{\pm 0.0211}$ | $0.1187_{\pm 0.0167}$ | $0.0785_{\pm 0.0018}$ | $\mathbf{0.0328}_{\pm 0.0042}$ |
| | 0.99 | $> 0.95$ | $0.7354_{\pm 0.0657}$ | $> 0.95$ | $> 0.95$ | $0.3835_{\pm 0.0032}$ | $\mathbf{0.1174}_{\pm 0.0127}$ |

### B.2.5 TENSOR-COMPLETION BASED RECOMMENDATION AS A DOWNSTREAM TASK

We split the MovieLens-100k dataset by time to form a tensor of size $943 \times 1682 \times 31$, where 31 denotes the number of weeks as a dimension of time. Since the baselines KBR-TC, TRLRF, OITNN-O, and the proposed $M^2$DMTF have high time cost on large tensors, we reduce the size of the tensor by removing the users giving 49 or less ratings and the movies rated by 49 or less users. Then we get a tensor of size $563 \times 593 \times 31$. The relative recovery error (average of 5 trials) are reported in Table 7. Our $M^2$DMTF outperformed other methods. In addition, comparing Table 7 with Table 1, we found that tensor completion methods are more effective than matrix completion methods in recommendation.

Table 7: Relative recovery errors of tensor completion methods on MovieLens-100k (MR denotes missing rate)

| MR | FaLRTC | TenALS | TMac | KBR-TC | TRLRF | CoSTCo | OITNN-O | $M^2$DMTF |
|---|---|---|---|---|---|---|---|---|
| 0.5 | 0.2627 | 0.2813 | 0.3065 | 0.2528 | 0.2634 | 0.2696 | 0.2830 | $\mathbf{0.2429}$ |
| 0.7 | 0.2692 | 0.3097 | 0.3470 | 0.2685 | 0.2719 | 0.2724 | 0.2951 | $\mathbf{0.2483}$ |
| 0.9 | 0.2881 | 0.3356 | 0.3782 | 0.2739 | 0.2804 | 0.2763 | 0.3207 | $\mathbf{0.2614}$ |

### B.2.6 VISUALIZATION OF FACTORS LEARNED BY $M^2$DMTF

Here we take the 'Flow injection' tensor (Nørgaard & Ridder, 1994) as an example. The data is from measuring samples with three different analytes on a flow injection analysis (FIA) system where a pH-gradient is imposed. The three analytes in twelve measured samples are 2-, 3-, and 4-hydroxy-benzaldehyde. They have different absorption spectra depending on whether they are in their acidic or basic form. The data set is a 12 (samples) × 100 (wavelengths) × 89 (times) array. Figure 9 presents the original data, incomplete data, and recovered data for the fifth horizontal slice (sample-5) of the 'Flow' tensor, where the missing rate is 0.97. We see that FaLRTC, TRLRF, and CoSTCo failed totally. TenALS, TMac, and KBR-TC have high recovery error visually. In contrast, the data recovered by our $M^2$DMTF is very close to the original data. More importantly, the result of $M^2$DMTF indicates that there are three analytes, while other methods failed to reveal the fact.

We analyze the characteristics of the low-dimensional factors learned by our $M^2$DMTF. Figure 10 presents the values of $\boldsymbol{U}_2 \in \mathbb{R}^{100 \times 20}$ and $\boldsymbol{Z}^{(2)} \in \mathbb{R}^{100 \times 10}$ as well as the correlation coefficients

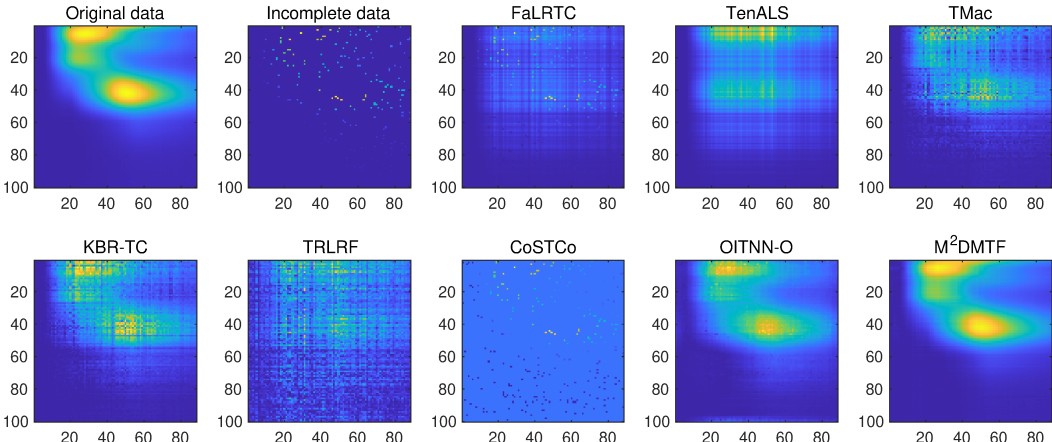

Figure 9: Completion result visualization of the fifth horizontal slice of the 'Flow' tensor (missing rate=0.97).

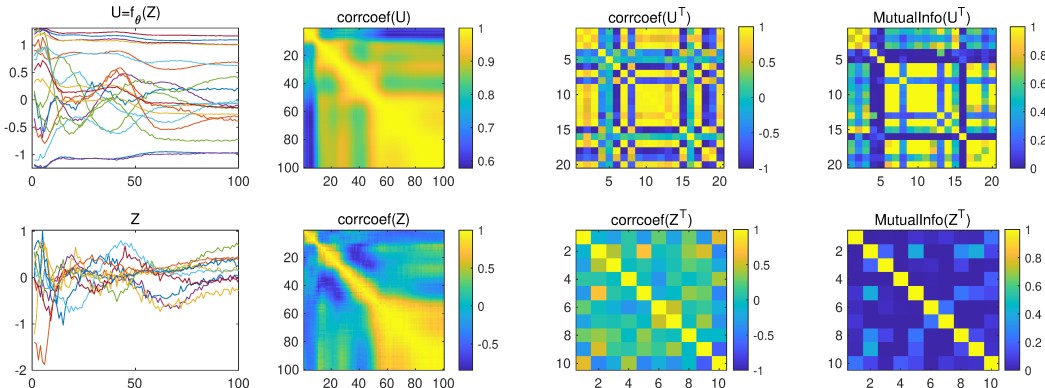

Figure 10: Latent factors $U_2$ and $Z^{(2)}$ (corresponding to the wavelengths dimension of 'Flow' tensor data) given by the proposed method M$^2$DMTF. Column 1: values of $U_2$ and $Z^{(2)}$. Column 2: correlation coefficients between the rows of $U_2$ and $Z^{(2)}$. Column 3: correlation coefficients between the columns of $U_2$ and $Z^{(2)}$. Column 4: mutual information between the columns of $U_2$ and $Z^{(2)}$.

and mutual information. In the first column, we see that the values of $U_2$ are more chaotic than the values $Z^{(2)}$. The reason is that the 20 variables in $U_2$ are correlated mutually. In the second column of Figure 10, we see that the autocorrelation of the samples (rows) of $U_2$ is similar to that of $Z^{(2)}$, because $U_2$ relies on $Z^{(2)}$. The third column of the figure indicates that the correlation between the variables in $U_2$ are higher than that in $Z^{(2)}$. The last column compares the mutual information that can quantify nonlinear relationship. Now comparing column 4 with column 3, we conclude that there exist significant nonlinear relationship between the variables in $U_2$ and it is necessary to represent $U_2$ by a lower-dimensional latent factor $Z^{(2)}$ via deep factorization. These results verified that the nonlinear latent factor assumption of our M$^2$DMTF is practical and useful.

## C    PROOF OF THEOREMS

First of all, we provide the following lemma.

**Lemma 1.** *Let $\mathcal{S}$ be a set defined over tensors of size $n_1 \times n_2 \times \cdots \times n_k$. Let $|\mathcal{S}|$ be the $\epsilon$-covering number of $\mathcal{S}$ with respect to the Frobenius norm. Let $N = \prod_{i=1}^{k} n_i$. Suppose $\boldsymbol{\mathcal{X}} \in \mathcal{S}$ and $\max\{\|\boldsymbol{\mathcal{Y}}\|_\infty, \|\boldsymbol{\mathcal{X}}\|_\infty\} \leq \xi$. Then with probability at least $1 - 2N^{-1}$, there exists a constant $c$ such*

*that*

$$\sup_{\boldsymbol{\mathcal{X}} \in \mathcal{S}} \left| \frac{1}{\sqrt{N}} \|\boldsymbol{\mathcal{Y}} - \boldsymbol{\mathcal{X}}\|_F - \frac{1}{\sqrt{|\Omega|}} \|\mathcal{P}_\Omega(\boldsymbol{\mathcal{Y}} - \boldsymbol{\mathcal{X}})\|_F \right| \leq \frac{2\epsilon}{\sqrt{|\Omega|}} + \left( \frac{8\xi^4 \log(|\mathcal{S}|N)}{|\Omega|} \right)^{1/4}.$$

The proof of the lemma is in Appendix D.1.

## C.1 Proof for Theorem 1

The following lemma (proved in Appendix 2) shows an upper bound for the covering number of the deep factorization matrix set.

**Lemma 2.** *Let $\mathcal{S} = \{\boldsymbol{X} \in \mathbb{R}^{m \times n} : \boldsymbol{X} = \boldsymbol{W}_L g(\boldsymbol{W}_{L-1} \cdots g(\boldsymbol{W}_1 \boldsymbol{Z})), \|\boldsymbol{Z}\|_F \leq \beta_0, \|\boldsymbol{W}_l\|_F \leq \beta_l, 1 \leq l \leq L\}$, where $\boldsymbol{Z} \in \mathbb{R}^{h_0 \times n}$, $\boldsymbol{W}_l \in \mathbb{R}^{h_l \times h_{l-1}}$, $l = 1, \ldots, L$, $h_L = m$, $h_0 = d$, and $h_{-1} = n$. Suppose the Lipschitz constant of $g$ is $\eta$. Then the covering numbers of $\mathcal{S}$ with respect to the Frobenius norm satisfy*

$$\mathcal{N}(\mathcal{S}, \|\cdot\|_F, \epsilon) \leq \left( \frac{3(L+1)\eta^{L-1} \prod_{l=0}^L \beta_l}{\epsilon} \right)^{\sum_{l=0}^L h_l h_{l-1}}.$$

Now using Lemma 1 and Lemma 2, we have

$$\frac{1}{\sqrt{N}} \|\boldsymbol{X} - \hat{\boldsymbol{X}}\|_F = \frac{1}{\sqrt{N}} \|\boldsymbol{Y} - \boldsymbol{E} - \hat{\boldsymbol{X}}\|_F$$

$$\leq \frac{1}{\sqrt{N}} \|\boldsymbol{Y} - \hat{\boldsymbol{X}}\|_F + \frac{1}{\sqrt{N}} \|\boldsymbol{E}\|_F$$

$$\leq \frac{1}{\sqrt{|\Omega|}} \|\mathcal{P}_\Omega(\boldsymbol{Y} - \hat{\boldsymbol{X}})\|_F + \frac{1}{\sqrt{N}} \|\boldsymbol{E}\|_F + \frac{2\epsilon}{\sqrt{|\Omega|}} + \left( \frac{8\xi^4 \log(|\mathcal{S}|N)}{|\Omega|} \right)^{1/4}$$

$$\leq \frac{1}{\sqrt{|\Omega|}} \|\mathcal{P}_\Omega(\boldsymbol{Y} - \hat{\boldsymbol{X}})\|_F + \frac{1}{\sqrt{N}} \|\boldsymbol{E}\|_F + \frac{2\epsilon}{\sqrt{|\Omega|}} + \left( \frac{8\xi^4 \left( \log N + \sum_{l=0}^L h_l h_{l-1} \log \frac{3(L+1)\eta^{L-1} \prod_{l=0}^L \beta_l}{\epsilon} \right)}{|\Omega|} \right)^{1/4}$$

$$\leq \frac{1}{\sqrt{|\Omega|}} \|\mathcal{P}_\Omega(\boldsymbol{Y} - \hat{\boldsymbol{X}})\|_F + \frac{1}{\sqrt{N}} \|\boldsymbol{E}\|_F + \frac{6\xi(L+1)}{\sqrt{|\Omega|}} + \left( \frac{8\xi^4 \left( \log N + \sum_{l=0}^L h_l h_{l-1} \log(\eta^{L-1}\xi^{-1} \prod_{l=0}^L \beta_l) \right)}{|\Omega|} \right)^{1/4}$$

$$\leq \frac{1}{\sqrt{|\Omega|}} \|\mathcal{P}_\Omega(\boldsymbol{Y} - \hat{\boldsymbol{X}})\|_F + \frac{1}{\sqrt{N}} \|\boldsymbol{E}\|_F + c\xi \left( \frac{\sum_{l=0}^L h_l h_{l-1} \log(\eta^{L-1}\xi^{-1} \prod_{l=0}^L \beta_l)}{|\Omega|} \right)^{1/4}.$$

In the fourth inequality, we have set $\epsilon = 3\xi(L+1)$. The last inequality holds because $N$ is much smaller than $|\mathcal{S}|$ and $c$ can be a constant large enough. This finished the proof.

## C.2 Proof for Theorem 2

The following lemma (proved in Appendix D.3) provides an upper bound for the covering number of the two-mode deep factorization matrix set.

**Lemma 3.** *Let $\mathcal{S} = \{\boldsymbol{X} \in \mathbb{R}^{n_1 \times n_2} : \boldsymbol{X} = f_{\theta_1}(\boldsymbol{Z})^\top \boldsymbol{Q} f_{\theta_2}(\boldsymbol{S}), \|\boldsymbol{Q}\|_F \leq \beta_Q, \|\boldsymbol{Z}\|_F \leq \beta_0^{(1)}, \|\boldsymbol{S}\|_F \leq \beta_0^{(2)}, \|\boldsymbol{W}_l^{(j)}\|_F \leq \beta_l^{(j)}, 1 \leq l \leq L_j, j = 1, 2; f_{\theta_1}(\boldsymbol{Z}) = g(\boldsymbol{W}_{L_1}^{(1)} g(\boldsymbol{W}_{L_1-1}^{(1)} \cdots g(\boldsymbol{W}_1^{(1)} \boldsymbol{Z}))), f_{\theta_2}(\boldsymbol{S}) = g(\boldsymbol{W}_{L_2}^{(2)} g(\boldsymbol{W}_{L_2-1}^{(2)} \cdots g(\boldsymbol{W}_1^{(2)} \boldsymbol{S}))), \}$, where $\boldsymbol{Z} \in \mathbb{R}^{d_1 \times n_1}$, $\boldsymbol{S} \in \mathbb{R}^{d_2 \times n_2}$, $\boldsymbol{W}_l^{(j)} \in \mathbb{R}^{h_l^{(j)} \times h_{l-1}^{(j)}}$, $l = 1, \ldots, L_j$, $h_{L_j}^{(j)} = m_j$, $h_0^{(j)} = d_j$, $h_{-1}^{(j)} = n_j$, $j = 1, 2$, and $\boldsymbol{Q} \in \mathbb{R}^{m_1 \times m_2}$. Suppose the Lipschitz constant of $g$ is $\eta$. Then the covering numbers of $\mathcal{S}$ with respect to the Frobenius norm satisfy*

$$\mathcal{N}(\mathcal{S}, \|\cdot\|_F, \epsilon) \leq \left( \frac{3(k+1)(\max(L_1, L_2)+1)s}{\epsilon} \right)^p,$$

*where $p = m_1 m_2 + \sum_{j=1}^2 \sum_{l=0}^{L_j} h_l^{(j)} h_{l-1}^{(j)}$ and $s = \beta_Q \eta^{L_1+L_2} \left( \prod_{l=0}^{L_1} \beta_l^{(1)} \right) \left( \prod_{l=0}^{L_2} \beta_l^{(2)} \right)$.*

Similar to the proof of Theorem 1, using Lemma 1, we have

$$
\frac{1}{\sqrt{N}}\|\boldsymbol{X} - \hat{\boldsymbol{X}}\|_F
$$

$$
\leq \frac{1}{\sqrt{|\Omega|}}\|\mathcal{P}_\Omega(\boldsymbol{Y} - \hat{\boldsymbol{X}})\|_F + \frac{1}{\sqrt{N}}\|\boldsymbol{E}\|_F + \frac{2\epsilon}{\sqrt{|\Omega|}} + \left(\frac{8\xi^4 \log\left(|\mathcal{S}|N\right)}{|\Omega|}\right)^{1/4},
$$

where $N = n_1 n_2$. Let $\epsilon = 3\xi(k+1)(\max(L_1, L_2)+1)$. According to Lemma 3 and we obtain

$$
\log(|\mathcal{S}|) = \left(m_1 m_2 + \sum_{j=1}^{2}\sum_{l=0}^{L_j} h_l^{(j)} h_{l-1}^{(j)}\right) \log\left(\beta_Q \eta^{L_1+L_2}\xi^{-1}(\prod_{l=0}^{L_1}\beta_l^{(1)})(\prod_{l=0}^{L_2}\beta_l^{(2)})\right).
$$

Now we have

$$
\frac{1}{\sqrt{N}}\|\boldsymbol{X} - \hat{\boldsymbol{X}}\|_F \leq \frac{1}{\sqrt{|\Omega|}}\|\mathcal{P}_\Omega(\boldsymbol{Y} - \hat{\boldsymbol{X}})\|_F + \frac{1}{\sqrt{N}}\|\boldsymbol{E}\|_F
$$
$$
+ c\xi\left(\frac{\left(m_1 m_2 + \sum_{j=1}^{2}\sum_{l=0}^{L_j} h_l^{(j)} h_{l-1}^{(j)}\right)\log\left(\beta_Q \eta^{L_1+L_2}\xi^{-1}\left(\prod_{l=0}^{L_1}\beta_l^{(1)}\right)\left(\prod_{l=0}^{L_2}\beta_l^{(2)}\right)\right)}{|\Omega|}\right)^{1/4}.
$$

This finished the proof.

### C.3  PROOF FOR THEOREM 3

The following lemma (proved in Appendix D.4) provides an upper bound for the covering number of the multi-mode deep factorization tensor set.

**Lemma 4.** *Let* $\mathcal{S} = \{\boldsymbol{\mathcal{X}} \in \mathbb{R}^{n_1 \times n_2 \cdots \times n_k} : \boldsymbol{\mathcal{X}} = \boldsymbol{\mathcal{C}} \times_1 f_{\theta_1}(\boldsymbol{Z}^{(1)}) \times_2 f_{\theta_2}(\boldsymbol{Z}^{(2)}) \cdots \times_k f_{\theta_k}(\boldsymbol{Z}^{(k)}); \ f_{\theta_j}(\boldsymbol{Z}) = g(\boldsymbol{W}_{L_j}^{(j)} g(\boldsymbol{W}_{L_j-1}^{(j)} \cdots g(\boldsymbol{W}_1^{(j)}\boldsymbol{Z}^{(j)}))), \ j = 1,\ldots,k; \ \|\boldsymbol{\mathcal{C}}\|_F \leq \beta_C, \|\boldsymbol{Z}^{(j)}\|_F \leq \beta_0^{(j)}, \|\boldsymbol{W}_l^{(j)}\|_F \leq \beta_l^{(j)}, 1 \leq l \leq L_j, j = 1,\ldots,k\}$, *where* $\boldsymbol{Z}^{(j)} \in \mathbb{R}^{d_j \times n_j}$, $\boldsymbol{W}_l^{(j)} \in \mathbb{R}^{h_l^{(j)} \times h_{l-1}^{(j)}}$, $l = 1,\ldots,L_j$, $h_{L_j}^{(j)} = m_j$, $h_0^{(j)} = d_j$, $h_{-1}^{(j)} = n_j$, $j = 1,\ldots,k$, *and* $\boldsymbol{\mathcal{C}} \in \mathbb{R}^{m_1 \times m_2 \cdots \times m_k}$. *Suppose the Lipschitz constant of* $g$ *is* $\eta$. *Then the covering numbers of* $\mathcal{S}$ *with respect to the Frobenius norm satisfy*

$$
\mathcal{N}(\mathcal{S}, \|\cdot\|_F, \epsilon) \leq \left(\frac{3(k+1)(L_{\max}+1)s}{\epsilon}\right)^p,
$$

*where* $p = \prod_{j=1}^{k} m_j + \sum_{j=1}^{k}\sum_{l=0}^{L_j} h_l^{(j)} h_{l-1}^{(j)}$ *and* $s = \beta_C \eta^{\sum_{j=1}^{k} L_j} \prod_{j=1}^{k}\prod_{l=0}^{L_j}\beta_l^{(j)}$.

The proof is similar to that for Theorem 2. Using Lemma 1 and Lemma 4 , we arrive at

$$
\frac{1}{\sqrt{N}}\|\boldsymbol{\mathcal{X}} - \hat{\boldsymbol{\mathcal{X}}}\|_F \leq \frac{1}{\sqrt{|\Omega|}}\|\mathcal{P}_\Omega(\boldsymbol{\mathcal{Y}} - \hat{\boldsymbol{\mathcal{X}}})\|_F + \frac{1}{\sqrt{N}}\|\boldsymbol{\mathcal{E}}\|_F
$$
$$
+ c\xi\left(\frac{\left(\prod_{j=1}^{k} m_j + \sum_{j=1}^{k}\sum_{l=0}^{L_j} h_l^{(j)} h_{l-1}^{(j)}\right)\log\left(\beta_C \eta^{\sum_{j=1}^{k} L_j}\xi^{-1}\prod_{j=1}^{k}\prod_{l=0}^{L_j}\beta_l^{(j)}\right)}{|\Omega|}\right)^{1/4},
$$

where $c$ is a constant.

## D  PROOF FOR LEMMAS

### D.1  PROOF FOR LEMMA 1

We give the following lemma.

**Lemma 5** (Hoeffding inequality for sampling without replacement (Serfling, 1974)). *Let $X_1, X_2, \ldots, X_s$ be a set of samples taken without replacement from a distribution $\{x_1, x_2, \ldots, x_N\}$ of mean $u$ and variance $\sigma^2$. Denote $a = \min_i x_i$ and $b = \max_i x_i$. Then*

$$P\left[\left|\frac{1}{s}\sum_{i=1}^{s} X_i - u\right| \geq t\right] \leq 2\exp\left(-\frac{2st^2}{(1-(s-1)/N)(b-a)^2}\right).$$

We define

$$\hat{\mathcal{L}}(\boldsymbol{\mathcal{X}}) := \frac{1}{|\Omega|}\|\mathcal{P}_\Omega(\boldsymbol{\mathcal{Y}} - \boldsymbol{\mathcal{X}})\|_F^2,$$

$$\mathcal{L}(\boldsymbol{\mathcal{X}}) := \frac{1}{N}\|\boldsymbol{\mathcal{Y}} - \boldsymbol{\mathcal{X}}\|_F^2,$$

where $N = \prod_{i=1}^{k} n_i$. Suppose $\max\{\|\boldsymbol{\mathcal{Y}}\|_\infty, \|\boldsymbol{\mathcal{X}}\|_\infty\} \leq \xi$. According to Lemma 5, we have

$$P\left[\hat{\mathcal{L}}(\boldsymbol{\mathcal{X}}) - \mathcal{L}(\boldsymbol{\mathcal{X}})| \geq t\right] \leq 2\exp\left(-\frac{2|\Omega|t^2}{(1-(|\Omega|-1)/N)\delta^2}\right),$$

where $\delta = 4\xi^2$. Using union bound for all $\bar{\boldsymbol{\mathcal{X}}} \in \mathcal{S}$ yields

$$P\left[\sup_{\bar{\boldsymbol{\mathcal{X}}}\in\mathcal{S}} |\hat{\mathcal{L}}(\bar{\boldsymbol{\mathcal{X}}}) - \mathcal{L}(\bar{\boldsymbol{\mathcal{X}}})| \geq t\right]$$

$$\leq 2|\mathcal{S}|\exp\left(-\frac{2|\Omega|t^2}{(1-(|\Omega|-1)/N)\delta^2}\right).$$

Or equivalently, with probability at least $1 - 2N^{-1}$,

$$\sup_{\bar{\boldsymbol{\mathcal{X}}}\in\mathcal{S}} |\hat{\mathcal{L}}(\bar{\boldsymbol{\mathcal{X}}}) - \mathcal{L}(\bar{\boldsymbol{\mathcal{X}}})| \leq \sqrt{\frac{\delta^2\log(|\mathcal{S}|N)}{2}\left(\frac{1}{|\Omega|} - \frac{1}{N} + \frac{1}{N|\Omega|}\right)}$$

$$\leq \sqrt{\frac{\delta^2\log(|\mathcal{S}|N)}{2|\Omega|}} \triangleq \Delta.$$

Since $|\sqrt{u} - \sqrt{v}| \leq \sqrt{|u-v|}$ holds for any non-negative $u$ and $v$, we have

$$\sup_{\bar{\boldsymbol{\mathcal{X}}}\in\mathcal{S}} \left|\sqrt{\hat{\mathcal{L}}(\bar{\boldsymbol{\mathcal{X}}})} - \sqrt{\mathcal{L}(\bar{\boldsymbol{\mathcal{X}}})}\right| \leq \sqrt{\Delta}.$$

Recall that $\epsilon \geq \|\boldsymbol{\mathcal{X}} - \bar{\boldsymbol{\mathcal{X}}}\|_F \geq \|\mathcal{P}(\boldsymbol{\mathcal{X}} - \bar{\boldsymbol{\mathcal{X}}})\|_F$, we have

$$\left|\sqrt{\mathcal{L}(\boldsymbol{\mathcal{X}})} - \sqrt{\mathcal{L}(\bar{\boldsymbol{\mathcal{X}}})}\right|$$

$$= \frac{1}{\sqrt{N}}\left|\|\boldsymbol{\mathcal{Y}} - \boldsymbol{\mathcal{X}}\|_F - \|\boldsymbol{\mathcal{Y}} - \bar{\boldsymbol{\mathcal{X}}}\|_F\right| \leq \frac{\epsilon}{\sqrt{N}}$$

and

$$\left|\sqrt{\hat{\mathcal{L}}(\boldsymbol{\mathcal{X}})} - \sqrt{\hat{\mathcal{L}}(\bar{\boldsymbol{\mathcal{X}}})}\right|$$

$$= \frac{1}{\sqrt{|\Omega|}}\left|\|\mathcal{P}_\Omega(\boldsymbol{\mathcal{Y}} - \boldsymbol{\mathcal{X}})\|_F - \|\mathcal{P}_\Omega(\boldsymbol{\mathcal{Y}} - \bar{\boldsymbol{\mathcal{X}}})\|_F\right| \leq \frac{\epsilon}{\sqrt{|\Omega|}}.$$

It follows that

$$\sup_{\boldsymbol{\mathcal{X}}\in\mathcal{S}} \left|\sqrt{\hat{\mathcal{L}}(\boldsymbol{\mathcal{X}})} - \sqrt{\mathcal{L}(\boldsymbol{\mathcal{X}})}\right|$$

$$\leq \sup_{\boldsymbol{\mathcal{X}}\in\mathcal{S}} \left|\sqrt{\hat{\mathcal{L}}(\boldsymbol{\mathcal{X}})} - \sqrt{\hat{\mathcal{L}}(\bar{\boldsymbol{\mathcal{X}}})}\right| + \left|\sqrt{\hat{\mathcal{L}}(\bar{\boldsymbol{\mathcal{X}}})} - \sqrt{\mathcal{L}(\bar{\boldsymbol{\mathcal{X}}})}\right| + \left|\sqrt{\mathcal{L}(\bar{\boldsymbol{\mathcal{X}}})} - \sqrt{\mathcal{L}(\boldsymbol{\mathcal{X}})}\right|$$

$$\leq \frac{\epsilon}{\sqrt{|\Omega|}} + \sqrt{\Delta} + \frac{\epsilon}{\sqrt{N}}$$

$$\leq \frac{2\epsilon}{\sqrt{|\Omega|}} + \left(\frac{\delta^2\log(|\mathcal{S}|N)}{2|\Omega|}\right)^{1/4}.$$

This finished the proof.

## D.2 PROOF FOR LEMMA 2

Let $\boldsymbol{X} = \boldsymbol{W}_L g(\boldsymbol{W}_{L-1} \cdots g(\boldsymbol{W}_1 \boldsymbol{Z}))$, where $\boldsymbol{Z} \in \mathbb{R}^{d \times n}$, $\boldsymbol{W}_l \in \mathbb{R}^{h_l \times h_{l-1}}$, $l = 1, \ldots, L$, and $h_0 = d$. Let $\mathcal{S}_{ab} := \{\boldsymbol{W} \in \mathbb{R}^{a \times b} : \|\boldsymbol{W}\|_F \leq \beta\}$. Then there exists an $\epsilon$-net $\bar{\mathcal{S}}_{ab}$ obeying

$$\mathcal{N}(\mathcal{S}_{ab}, \|\cdot\|_F, \epsilon) \leq \left(\frac{3\beta}{\epsilon}\right)^{ab}$$

such that $\|\boldsymbol{W} - \bar{\boldsymbol{W}}\|_F \leq \epsilon$. Now replace $\epsilon$ with $\epsilon/\gamma$ and let $\|\boldsymbol{W}_l - \bar{\boldsymbol{W}}_l\|_F \leq \dfrac{\epsilon}{\gamma_l}$, $l = 1, \ldots, L$,

$\|\boldsymbol{Z} - \bar{\boldsymbol{Z}}\|_F \leq \dfrac{\epsilon}{\gamma_0}$. Let $\gamma_l = (L+1)\eta^{L-1} \prod_{i \neq l} \beta_i$, $l = 0, \ldots, L$.

$$\begin{aligned}
&\|\boldsymbol{X} - \bar{\boldsymbol{X}}\|_F \\
=&\|\boldsymbol{W}_L g(\boldsymbol{W}_{L-1} \cdots \boldsymbol{W}_2 g(\boldsymbol{W}_1 \boldsymbol{Z})) - \bar{\boldsymbol{W}}_L g(\bar{\boldsymbol{W}}_{L-1} \cdots \bar{\boldsymbol{W}}_2 g(\bar{\boldsymbol{W}}_1 \bar{\boldsymbol{Z}}))\|_F \\
=&\|\boldsymbol{W}_L g(\boldsymbol{W}_{L-1} \cdots \boldsymbol{W}_2 g(\boldsymbol{W}_1 \boldsymbol{Z})) \pm \bar{\boldsymbol{W}}_L g(\boldsymbol{W}_{L-1} \cdots \boldsymbol{W}_2 g(\boldsymbol{W}_1 \boldsymbol{Z})) \\
&\pm \bar{\boldsymbol{W}}_L g(\bar{\boldsymbol{W}}_{L-1} \cdots \boldsymbol{W}_2 g(\boldsymbol{W}_1 \boldsymbol{Z})) \cdots \pm \bar{\boldsymbol{W}}_L g(\bar{\boldsymbol{W}}_{L-1} \cdots \bar{\boldsymbol{W}}_2 g(\boldsymbol{W}_1 \boldsymbol{Z})) \\
&\pm \bar{\boldsymbol{W}}_L g(\bar{\boldsymbol{W}}_{L-1} \cdots \bar{\boldsymbol{W}}_2 g(\bar{\boldsymbol{W}}_1 \boldsymbol{Z})) - \bar{\boldsymbol{W}}_L g(\bar{\boldsymbol{W}}_{L-1} \cdots g(\bar{\boldsymbol{W}}_1 \bar{\boldsymbol{Z}}))\|_F \\
\leq&\|\boldsymbol{W}_L g(\boldsymbol{W}_{L-1} \cdots \boldsymbol{W}_2 g(\boldsymbol{W}_1 \boldsymbol{Z})) - \bar{\boldsymbol{W}}_L g(\boldsymbol{W}_{L-1} \cdots \boldsymbol{W}_2 g(\boldsymbol{W}_1 \boldsymbol{Z}))\|_F \\
&+ \|\bar{\boldsymbol{W}}_L g(\boldsymbol{W}_{L-1} \cdots \boldsymbol{W}_2 g(\boldsymbol{W}_1 \boldsymbol{Z})) - \bar{\boldsymbol{W}}_L g(\bar{\boldsymbol{W}}_{L-1} \cdots \boldsymbol{W}_2 g(\boldsymbol{W}_1 \boldsymbol{Z}))\|_F \\
&+ \cdots + \|\bar{\boldsymbol{W}}_L g(\bar{\boldsymbol{W}}_{L-1} \cdots \bar{\boldsymbol{W}}_2 g(\boldsymbol{W}_1 \boldsymbol{Z})) - \bar{\boldsymbol{W}}_L g(\bar{\boldsymbol{W}}_{L-1} \cdots \bar{\boldsymbol{W}}_2 g(\bar{\boldsymbol{W}}_1 \boldsymbol{Z}))\|_F \\
&+ \|\bar{\boldsymbol{W}}_L g(\bar{\boldsymbol{W}}_{L-1} \cdots \bar{\boldsymbol{W}}_2 g(\bar{\boldsymbol{W}}_1 \boldsymbol{Z})) - \bar{\boldsymbol{W}}_L g(\bar{\boldsymbol{W}}_{L-1} \cdots \bar{\boldsymbol{W}}_2 g(\bar{\boldsymbol{W}}_1 \bar{\boldsymbol{Z}}))\|_F \\
\leq&\eta^{L-1}\|\boldsymbol{W}_L - \bar{\boldsymbol{W}}_L\|_F \|\boldsymbol{Z}\|_F \prod_{l=1}^{L-1} \|\boldsymbol{W}_l\|_F \\
&+ \eta^{L-1}\|\boldsymbol{W}_{L-1} - \bar{\boldsymbol{W}}_{L-1}\|_F \|\boldsymbol{Z}\|_F \|\bar{\boldsymbol{W}}_L\|_F \prod_{l=1}^{L-2} \|\boldsymbol{W}_l\|_F \\
&+ \cdots + \eta^{L-1}\|\boldsymbol{W}_1 - \bar{\boldsymbol{W}}_1\|_F \|\boldsymbol{Z}\|_F \prod_{l=2}^{L-1} \|\bar{\boldsymbol{W}}_l\|_F \\
&+ \eta^{L-1}\|\boldsymbol{Z} - \bar{\boldsymbol{Z}}\|_F \prod_{l=1}^{L} \|\bar{\boldsymbol{W}}_l\|_F \\
\leq&\eta^{L-1}\frac{\epsilon}{\gamma_L} \prod_{l=0}^{L-1} \beta_l + \eta^{L-1}\frac{\epsilon}{\gamma_{L-1}} \prod_{l \neq L-1} \beta_l + \cdots + \eta^{L-1}\frac{\epsilon}{\gamma_1} \prod_{l \neq 1} \beta_l + \eta^{L-1}\frac{\epsilon}{\gamma_0} \prod_{l \neq 0} \beta_l \\
=&\epsilon.
\end{aligned} \tag{19}$$

The second inequality utilized the Lipschitz continuity of $g$ and the submultiplicativity of the Frobenius norm. Therefore, $\bar{\mathcal{S}}$ is an $\epsilon$-cover of $\mathcal{S}$. We have

$$\begin{aligned}
&\mathcal{N}(\mathcal{S}, \|\cdot\|_F, \epsilon) \\
\leq&\prod_{l=0}^{L} \left(\frac{3\beta_l \gamma_l}{\epsilon}\right)^{h_l h_{l-1}} \\
=&\prod_{l=0}^{L} \left(\frac{3(L+1)\eta^{L-1} \prod_{l=0}^{L} \beta_l}{\epsilon}\right)^{h_l h_{l-1}} \\
=&\left(\frac{3(L+1)\eta^{L-1} \prod_{l=0}^{L} \beta_l}{\epsilon}\right)^{\sum_{l=0}^{L} h_l h_{l-1}}.
\end{aligned}$$

## D.3 PROOF FOR LEMMA 3

Lemma 3 is a special case of Lemma 4.

### D.4 Proof for Lemma 4

Let $\rho_0 = \|\boldsymbol{\mathcal{C}}\|_F$. For $j = 1, \ldots, k$, we have

$$\|\boldsymbol{U}_j\|_F \le \eta^{L_j} \|\boldsymbol{Z}^{(j)}\|_F \prod_{l=1}^{L_j} \|\boldsymbol{W}_l^{(j)}\|_F = \eta^{L_j} \prod_{l=0}^{L_j} \beta_l^{(j)} \triangleq \rho_j. \tag{20}$$

Denote $\tau = \prod_{j=0}^{k} \rho_j$. Similar values can be defined for $\bar{\boldsymbol{\mathcal{C}}}$ and $\bar{\boldsymbol{U}}_j$.

Let $\mathcal{S}_{jl} := \{\boldsymbol{W}_l^{(j)} \in \mathbb{R}^{a \times b} : \|\boldsymbol{W}_l^{(j)}\|_F \le \beta_l^{(j)}\}$. Then there exists an $\epsilon$-net $\bar{\mathcal{S}}_{jl}$ obeying

$$\mathcal{N}(\mathcal{S}_{jl}, \|\cdot\|_F, \epsilon) \le \left(\frac{3\beta_l^{(j)}}{\epsilon}\right)^{h_l^{(j)} h_{l-1}^{(j)}}$$

such that $\|\boldsymbol{W}_l^{(j)} - \bar{\boldsymbol{W}}_l^{(j)}\|_F \le \epsilon$. Let $\Gamma_l^{(j)} = (L_j + 1)\eta^{L_j} \prod_{i \ne l} \beta_i$, $l = 0, \ldots, L_j$, and $\Upsilon_j = (k+1)\prod_{t \ne j} \rho_t$, $j = 0, 1, \ldots, k$. Replace $\epsilon$ with $\frac{\epsilon}{\Gamma_l^{(j)} \Upsilon_j}$, we have $\|\boldsymbol{W}_l^{(j)} - \bar{\boldsymbol{W}}_l^{(j)}\|_F \le \frac{\epsilon}{\Gamma_l^{(j)} \Upsilon_j}$ and have

$$\mathcal{N}(\mathcal{S}_{jl}, \|\cdot\|_F, \tfrac{\epsilon}{\Gamma_l^{(j)} \Upsilon_j}) \le \left(\frac{3\Gamma_l^{(j)} \Upsilon_j \beta_l^{(j)}}{\epsilon}\right)^{h_l^{(j)} h_{l-1}^{(j)}}.$$

It follows that

$$\|\boldsymbol{U}_j - \bar{\boldsymbol{U}}_j\|_F$$
$$= \|g(\boldsymbol{W}_{L_j}^{(j)} g(\boldsymbol{W}_{L_j-1}^{(j)} \cdots \boldsymbol{W}_2^{(j)} g(\boldsymbol{W}_1^{(j)} \boldsymbol{Z}^{(j)}))) - g(\bar{\boldsymbol{W}}_{L_j}^{(j)} g(\bar{\boldsymbol{W}}_{L_j-1}^{(j)} \cdots \bar{\boldsymbol{W}}_2^{(j)} g(\bar{\boldsymbol{W}}_1^{(j)} \bar{\boldsymbol{Z}}^{(j)})))\|_F$$
$$= \|g(\boldsymbol{W}_{L_j}^{(j)} g(\boldsymbol{W}_{L_j-1}^{(j)} \cdots \boldsymbol{W}_2^{(j)} g(\boldsymbol{W}_1^{(j)} \boldsymbol{Z}^{(j)}))) \pm g(\bar{\boldsymbol{W}}_{L_j}^{(j)} g(\boldsymbol{W}_{L_j-1}^{(j)} \cdots \boldsymbol{W}_2^{(j)} g(\boldsymbol{W}_1^{(j)} \bar{\boldsymbol{Z}}^{(j)})))$$
$$\pm g(\bar{\boldsymbol{W}}_{L_j}^{(j)} g(\bar{\boldsymbol{W}}_{L_j-1}^{(j)} \cdots \boldsymbol{W}_2^{(j)} g(\boldsymbol{W}_1^{(j)} \boldsymbol{Z}^{(j)}))) \cdots \pm g(\bar{\boldsymbol{W}}_{L_j}^{(j)} g(\bar{\boldsymbol{W}}_{L_j-1}^{(j)} \cdots \bar{\boldsymbol{W}}_2^{(j)} g(\boldsymbol{W}_1^{(j)} \boldsymbol{Z}^{(j)})))$$
$$\pm g(\bar{\boldsymbol{W}}_L g(\bar{\boldsymbol{W}}_{L-1} \cdots \bar{\boldsymbol{W}}_2 g(\bar{\boldsymbol{W}}_1 \boldsymbol{Z})) - g(\bar{\boldsymbol{W}}_{L_j}^{(j)} g(\bar{\boldsymbol{W}}_{L_j-1}^{(j)} \cdots g(\bar{\boldsymbol{W}}_1^{(j)} \bar{\boldsymbol{Z}}^{(j)})))\|_F$$
$$\le \|g(\boldsymbol{W}_{L_j} g(\boldsymbol{W}_{L_j-1}^{(j)} \cdots \boldsymbol{W}_2^{(j)} g(\boldsymbol{W}_1^{(j)} \boldsymbol{Z}^{(j)}))) - g(\bar{\boldsymbol{W}}_{L_j}^{(j)} g(\boldsymbol{W}_{L_j-1}^{(j)} \cdots \boldsymbol{W}_2^{(j)} g(\boldsymbol{W}_1^{(j)} \boldsymbol{Z}^{(j)})))\|_F$$
$$+ \|g(\bar{\boldsymbol{W}}_{L_j}^{(j)} g(\boldsymbol{W}_{L_j-1}^{(j)} \cdots \boldsymbol{W}_2^{(j)} g(\boldsymbol{W}_1^{(j)} \boldsymbol{Z}^{(j)}))) - g(\bar{\boldsymbol{W}}_{L_j}^{(j)} g(\bar{\boldsymbol{W}}_{L_j-1}^{(j)} \cdots \boldsymbol{W}_2^{(j)} g(\boldsymbol{W}_1^{(j)} \boldsymbol{Z}_j)))\|_F$$
$$+ \cdots + \|g(\bar{\boldsymbol{W}}_{L_j}^{(j)} g(\bar{\boldsymbol{W}}_{L_j-1}^{(j)} \cdots \bar{\boldsymbol{W}}_2^{(j)} g(\boldsymbol{W}_1^{(j)} \boldsymbol{Z}^{(j)}))) - g(\bar{\boldsymbol{W}}_{L_j}^{(j)} g(\bar{\boldsymbol{W}}_{L_j-1}^{(j)} \cdots \bar{\boldsymbol{W}}_2^{(j)} g(\bar{\boldsymbol{W}}_1^{(j)} \boldsymbol{Z}_j)))\|_F$$
$$+ \|g(\bar{\boldsymbol{W}}_{L_j}^{(j)} g(\bar{\boldsymbol{W}}_{L_j-1}^{(j)} \cdots \bar{\boldsymbol{W}}_2^{(j)} g(\bar{\boldsymbol{W}}_1^{(j)} \boldsymbol{Z}^{(j)}))) - g(\bar{\boldsymbol{W}}_{L_j}^{(j)} g(\bar{\boldsymbol{W}}_{L_j-1}^{(j)} \cdots \bar{\boldsymbol{W}}_2^{(j)} g(\bar{\boldsymbol{W}}_1^{(j)} \bar{\boldsymbol{Z}}^{(j)})))\|_F$$
$$\le \eta^{L_j} \|\boldsymbol{W}_{L_j}^{(j)} - \bar{\boldsymbol{W}}_{L_j}^{(j)}\|_F \|\boldsymbol{Z}^{(j)}\|_F \prod_{l=1}^{L_j-1} \|\boldsymbol{W}_l^{(j)}\|_F$$
$$+ \eta^{L_j} \|\boldsymbol{W}_{L_j-1}^{(j)} - \bar{\boldsymbol{W}}_{L_j-1}^{(j)}\|_F \|\boldsymbol{Z}^{(j)}\|_F \|\bar{\boldsymbol{W}}_{L_j}^{(j)}\|_F \prod_{l=1}^{L_j-2} \|\boldsymbol{W}_l^{(j)}\|_F$$
$$+ \cdots + \eta^{L_j} \|\boldsymbol{W}_1^{(j)} - \bar{\boldsymbol{W}}_1^{(j)}\|_F \|\boldsymbol{Z}^{(j)}\|_F \prod_{l=2}^{L-1} \|\bar{\boldsymbol{W}}_l^{(j)}\|_F$$
$$+ \eta^{L_j} \|\boldsymbol{Z}^{(j)} - \bar{\boldsymbol{Z}}^{(j)}\|_F \prod_{l=1}^{L_j} \|\bar{\boldsymbol{W}}_l^{(j)}\|_F$$
$$\le \eta^{L_j} \frac{\epsilon}{\Gamma_L^{(j)} \Upsilon_j} \prod_{l=0}^{L_j-1} \beta_l^{(j)} + \eta^{L_j} \frac{\epsilon}{\Gamma_{L-1}^{(j)} \Upsilon_j} \prod_{l \ne L_j-1} \beta_l^{(j)} + \cdots + \eta^{L_j} \frac{\epsilon}{\Gamma_1^{(j)} \Upsilon_j} \prod_{l \ne 1} \beta_l^{(j)} + \eta^{L_j} \frac{\epsilon}{\Gamma_0^{(j)} \Upsilon_j} \prod_{l \ne 0} \beta_l^{(j)}$$
$$= \frac{\epsilon}{\Upsilon_j}. \tag{21}$$

Now we have

$$
\begin{aligned}
&\|\boldsymbol{\mathcal{X}} - \bar{\boldsymbol{\mathcal{X}}}\|_F \\
=&\|\boldsymbol{\mathcal{C}} \times_1 \boldsymbol{U}_1 \times_2 \cdots \times_k \boldsymbol{U}_k - \bar{\boldsymbol{\mathcal{C}}} \times_1 \bar{\boldsymbol{U}}_1 \times_2 \cdots \times_k \bar{\boldsymbol{U}}_k\|_F \\
=&\|\boldsymbol{\mathcal{C}} \times_1 \boldsymbol{U}_1 \times_2 \cdots \times_k \boldsymbol{U}_k \pm \boldsymbol{\mathcal{C}} \times_1 \boldsymbol{U}_1 \times_2 \cdots \times_k \bar{\boldsymbol{U}}_k \pm \boldsymbol{\mathcal{C}} \times_1 \boldsymbol{U}_1 \times_2 \cdots \times_{k-1} \bar{\boldsymbol{U}}_{k-1} \times_k \bar{\boldsymbol{U}}_k \\
&\pm \cdots \pm \boldsymbol{\mathcal{C}} \times_1 \bar{\boldsymbol{U}}_1 \times_2 \cdots \times_{k-1} \bar{\boldsymbol{U}}_{k-1} \times_k \bar{\boldsymbol{U}}_k - \bar{\boldsymbol{\mathcal{C}}} \times_1 \bar{\boldsymbol{U}}_1 \times_2 \cdots \times_k \bar{\boldsymbol{U}}_k\|_F \\
\leq&\|\boldsymbol{\mathcal{C}} \times_1 \boldsymbol{U}_1 \times_2 \cdots \times_k (\boldsymbol{U}_k - \bar{\boldsymbol{U}}_k)\|_F + \|\boldsymbol{\mathcal{C}} \times_1 \boldsymbol{U}_1 \times_2 \cdots \times_{k-1} (\boldsymbol{U}_{k-1} - \bar{\boldsymbol{U}}_{k-1}) \times_k \bar{\boldsymbol{U}}_k\|_F \\
&+ \cdots + \|\boldsymbol{\mathcal{C}} \times_1 (\boldsymbol{U}_1 - \bar{\boldsymbol{U}}_1) \times_2 \bar{\boldsymbol{U}}_2 \cdots \times_k \bar{\boldsymbol{U}}_k\|_F + \|(\boldsymbol{\mathcal{C}} - \bar{\boldsymbol{\mathcal{C}}}) \times_1 \bar{\boldsymbol{U}}_1 \times_2 \bar{\boldsymbol{U}}_2 \cdots \times_k \bar{\boldsymbol{U}}_k\|_F \\
\leq&\|\boldsymbol{\mathcal{C}}\|_F \|\boldsymbol{U}_1\|_F \cdots \|\boldsymbol{U}_{k-1}\|_F \|\boldsymbol{U}_k - \bar{\boldsymbol{U}}_k\|_F + \|\boldsymbol{\mathcal{C}}\|_F \|\boldsymbol{U}_1\|_F \cdots \|\boldsymbol{U}_{k-1} - \bar{\boldsymbol{U}}_{k-1}\|_F \|\bar{\boldsymbol{U}}_k\|_F \\
&+ \cdots + \|\boldsymbol{\mathcal{C}}\|_F \|\boldsymbol{U}_1 - \bar{\boldsymbol{U}}_1\|_F \|\bar{\boldsymbol{U}}_2\|_F \cdots \|\bar{\boldsymbol{U}}_k\|_F + \|\boldsymbol{\mathcal{C}} - \bar{\boldsymbol{\mathcal{C}}}\|_F \|\bar{\boldsymbol{U}}_1\|_F \|\bar{\boldsymbol{U}}_2\|_F \cdots \|\bar{\boldsymbol{U}}_k\|_F \\
\leq&\|\boldsymbol{\mathcal{C}}\|_F \|\boldsymbol{U}_1\|_F \cdots \|\boldsymbol{U}_{k-1}\|_F \frac{\epsilon}{\Upsilon_k} + \|\boldsymbol{\mathcal{C}}\|_F \|\boldsymbol{U}_1\|_F \cdots \frac{\epsilon}{\Upsilon_{k-1}} \|\bar{\boldsymbol{U}}_k\|_F \\
&+ \cdots + \|\boldsymbol{\mathcal{C}}\|_F \frac{\epsilon}{\Upsilon_1} \|\bar{\boldsymbol{U}}_2\|_F \cdots \|\bar{\boldsymbol{U}}^k\|_F + \frac{\epsilon}{\Upsilon_0} \|\bar{\boldsymbol{U}}_1\|_F \|\bar{\boldsymbol{U}}_2\|_F \cdots \|\bar{\boldsymbol{U}}^k\|_F \\
\leq&\epsilon \left( \frac{1}{\Upsilon_k} \prod_{j \neq k} \rho_j + \frac{1}{\Upsilon_{k-1}} \prod_{j \neq k-1} \rho_j + \cdots + \frac{1}{\Upsilon_1} \prod_{j \neq 1} \rho_j + \frac{1}{\Upsilon_0} \prod_{j \neq 0} \rho_j \right) \\
=&\epsilon
\end{aligned}
$$

Therefore $\bar{\mathcal{S}}$ is an $\epsilon$-cover of $\mathcal{S}$. Then the covering number of $\mathcal{S}_{dnk}$ can be bounded as

$$
\begin{aligned}
&\mathcal{N}(\mathcal{S}_{dnk}, \|\cdot\|_F, \epsilon) \\
\leq& \left( \frac{3\Upsilon_0 \rho_0}{\epsilon} \right)^{\prod_{j=1}^k m_j} \prod_{j=1}^k \prod_{l=0}^{L_j} \left( \frac{3\Gamma_l^{(j)} \Upsilon_j \beta_l^{(j)}}{\epsilon} \right)^{h_l^{(j)} h_{l-1}^{(j)}} . \\
=& \left( \frac{3\Upsilon_0 \rho_0}{\epsilon} \right)^{\prod_{j=1}^k m_j} \prod_{j=1}^k \prod_{l=0}^{L_j} \left( \frac{3(L_j + 1)\rho_j \Upsilon_j}{\epsilon} \right)^{h_l^{(j)} h_{l-1}^{(j)}} \\
=& \left( \frac{3(k+1)\tau}{\epsilon} \right)^{\prod_{j=1}^k m_j} \prod_{j=1}^k \prod_{l=0}^{L_j} \left( \frac{3(k+1)(L_j + 1)\tau}{\epsilon} \right)^{h_l^{(j)} h_{l-1}^{(j)}} . \\
\leq& \left( \frac{3(k+1)\tau}{\epsilon} \right)^{\prod_{j=1}^k m_j} \prod_{j=1}^k \left( \frac{3(k+1)(L_{\max} + 1)\tau}{\epsilon} \right)^{\sum_{l=0}^{L_j} h_l^{(j)} h_{l-1}^{(j)}} \\
=& \left( \frac{3(k+1)\tau}{\epsilon} \right)^{\prod_{j=1}^k m_j} \left( \frac{3(k+1)(L_{\max} + 1)\tau}{\epsilon} \right)^{\sum_{j=1}^k \sum_{l=0}^{L_j} h_l^{(j)} h_{l-1}^{(j)}} \\
\leq& \left( \frac{3(k+1)(L_{\max} + 1)\tau}{\epsilon} \right)^{\prod_{j=1}^k m_j + \sum_{j=1}^k \sum_{l=0}^{L_j} h_l^{(j)} h_{l-1}^{(j)}} \\
=& \left( \frac{3(k+1)(L_{\max} + 1)\rho_0 \prod_{j=1}^k \left( \eta^{L_j} \prod_{l=0}^{L_j} \beta_l^{(j)} \right)}{\epsilon} \right)^{\prod_{j=1}^k m_j + \sum_{j=1}^k \sum_{l=0}^{L_j} h_l^{(j)} h_{l-1}^{(j)}} .
\end{aligned}
$$

