# OpenReview forum: "Multi-Mode Deep Matrix and Tensor Factorization"
_ICLR.cc/2022/Conference — ICLR 2022 Poster_

### Official Review · Reviewer_BPJG · 2021-11-02

**Correctness:** 3
**Technical Novelty And Significance:** 3
**Empirical Novelty And Significance:** 3
**Recommendation:** 8
**Confidence:** 2

**Main Review:**

The paper discusses why and when the nonlinear DMF outperform linear MF, which is very important in practice. The proposed two-mode DMF method and its tensor variant are novel and interesting. This paper provides comprehensive experiments, including both synthetic and real matrix/tensor data sets, to show the effectiveness of the proposed framework. The results based on various optimization algorithms make the performance convincing and the comparison fair.

In addition, there are some other comments to enhance the quality of the paper:
1. In eqn.(2), it that a typo for the half parenthesis in the last term?

2. It is quite confusing for the symbol "h", which sometimes denotes the dimension, e.g., in (5), while sometimes represents the function in p.5. It may be better to distinguish them with different notation.

3. In line -9 p.3, does it mean that one particular $h_l$ is large or all $h_l$'s are large?

4. The Assumption 2 in Sec 3 is not well organized. First, the definitions for $f_{\theta_i}$ could be moved to eqn.(8) when they first appeared. Second, should the functions $f_{\theta_i}$ for $i=1,2$ defined from one matrix space to another, instead of $\mathbb{R}^{d_i}\to\mathbb{R}^{m_i}$? Also, the mapping notation here should be $\to$ rather than $\mapsto$. Lastly, the parameter spaces are not clearly defined, e.g., the spaces where the parameters $\theta_1$ and $\theta_2$ belong to.

5. Minor issues:
line 5 p.4: What does it mean by "d can be larger than the ground truth"? It seems that d is the dimension of the latent variable.
line -14 p.5: "tighter generalization bound" -> "a tighter generalization bound"
line -10 p.5: "could be" -> "are"
line 11 p.7: "higher-order" -> "high-order"?


**Summary Of The Paper:**

The paper provides a multi-mode framework for the deep learning based tensor decomposition, which could be useful for dealing with nonlinear high-dimensional data sets. In particular, it extends the deep matrix factorization (DMF) method and proposes a multi-mode deep matrix factorization method for matrix completion with convergence guarantee. Based on this, it also develops a multi-mode nonlinear deep tensor factorization method with convergence guarantee. The proposed models are solved by various optimization algorithms. Numerical experiments on synthetic and real data sets of the matrix/tensor form have shown that the proposed methods outperform other state of the arts.

**Summary Of The Review:**

Overall, I vote for acceptance. The multi-mode generalization of deep matrix factorization and its extension to tensors are insightful with detailed theoretical discussions. My major concern is about the clarity of the paper and some notational confusion. Hopefully, the authors can address my concern in the rebuttal period.

---

> ### Author Response · Authors · 2021-11-19
> **Detailed response**
>
> Thank you so much for recognizing our contribution. Here we repose to your questions one by one.
>
> $\textbf{Q1:}$ In eqn.(2), is that a typo for the half parenthesis in the last term?
>
> $\textbf{Response: }$ Yes. We deleted it in the revised paper.
>
> $\textbf{Q2:}$ It is quite confusing for the symbol "h", which sometimes denotes the dimension, e.g., in (5), while sometimes represents the function in p.5. It may be better to distinguish them with different notation.
>
> $\textbf{Response: }$ We are sorry for the confusing notation and thank you for pointing out.  We replaced the "h" in page 5 with 'R", to denote "regularization".
>
> $\textbf{Q3:}$ In line -9 p.3, does it mean that one particular $h_l$ is large or all $h_l$'s are large?
>
> $\textbf{Response: }$ It means there exists one large enough $h_l$. Because according to the universal approximation theorems, a neural network with one hidden layer can approximate any continuous functions under some mild conditions provided that the layer is sufficiently wide, i.e., that $h_l$ is sufficiently large.   In the revised paper, we modified the sentence to
> "According to the universal approximation theorems (Cybenko, 1989; Pinkus, 1999; Lu et al., 2017; Hanin and Sellke, 2017),  $f$ can be well approximated by the neural network provided that $L$ is sufficiently large or at least one of $h_1,h_2,\ldots,h_{L-1}$ is sufficiently large."
>
>
> $\textbf{Q3:}$ The Assumption 2 in Sec 3 is not well organized. First, the definitions for $f_{\theta_i}$ could be moved to eqn.(8) when they first appeared. Second, should the functions $f_{\theta_i}$ for $i=1,2$ defined from one matrix space to another, instead of $\mathbb{R}^{d_i}\rightarrow\mathbb{R}^{m_i}$? Also, the mapping notation here should be $\rightarrow$ rather than $\mapsto$. Lastly, the parameter spaces are not clearly defined, e.g., the spaces where the parameters  $\theta_1$ and $\theta_2$  belong to.
>
>
> $\textbf{Response: }$
>
> 1) Thank you very much for the suggestions. In the revised paper, we moved the definitions for $f _ {\theta_i}$ to eqn.(8). Namely,  following eqn.(8), we state:
>
> "In (8), $f_{\theta_1}:\mathbb{R}^{d_1}\rightarrow\mathbb{R}^{m_1}$ and $f _ {\theta_2}:\mathbb{R}^{d_2}\rightarrow\mathbb{R}^{m_2}$ are multi-layer neural networks with parameter sets
>  $\theta _ 1= \lbrace\mathbf{W} _ {1} ^{(1)}, \ldots, \mathbf{W} _ {L_1} ^{(1)}\rbrace$ and $\theta _ 2= \lbrace\mathbf{W} _ {1} ^{(2)}, \ldots, \mathbf{W} _ {L_2} ^{(2)}\rbrace$, where $\mathbf{W}_{i}^{(j)}$ denotes the weight matrix in the $i$-th layer of neural network $j$,  and $L_j$ denotes the number of layers of neural network $j$,  $j=1,2$. "
>
> 2) The functions $f_{\theta_i}$ for $i=1,2$ cannot be defined from one matrix space to another because we are considering the nonlinear latent variable model in vector spaces, i.e., the latent variable and the output are both vectors.  In $\hat{ \mathbf{U} } = f_{\theta_1} (\hat{\mathbf{Z}}) $ and $ \hat{\mathbf{V}}=f_{\theta_2}(\hat{\mathbf{S}})$, $f_{\theta_1}$ and $f_{\theta_2}$ are performed on each column of $\hat{\mathbf{Z}}$ and $\hat{\mathbf{S}}$ respectively.  Sorry for the confusing definition.  In the revised paper, we have rewritten (8) as
>
>
> $$\qquad\mathbf{Y}\approx \hat{\mathbf{U}}^T\hat{\mathbf{Q}}\hat{\mathbf{V}} ,\quad \hat{\mathbf{u}}_{i_1} = f _ {\theta _1} (\hat{\mathbf{z}} _ {i_1}),\quad \hat{\mathbf{v}} _ {i_2}= f _ {\theta_2} ( \hat{\mathbf{s} } _ {i_2}), ~~~i_1=1,\ldots,n_1,~~~i_2=1,\ldots,n_2.$$
>
>
>
> 3) We corrected the mapping notation. Thanks for pointing out.
>
> 4) In the revised paper, we explicitly defined $\theta_1$ and $\theta_2$ as
> $\theta _ 1= \lbrace\mathbf{W} _ {1} ^{(1)}, \ldots, \mathbf{W} _ {L_1} ^{(1)}\rbrace$ and $\theta _ 2= \lbrace\mathbf{W} _ {1} ^{(2)}, \ldots, \mathbf{W} _ {L_2} ^{(2)}\rbrace$.
>
>
>
> $\textbf{Q3:}$ Minor issues: line 5 p.4: What does it mean by "d can be larger than the ground truth"? It seems that d is the dimension of the latent variable.  line -14 p.5: "tighter generalization bound" -> "a tighter generalization bound" line -10 p.5: "could be" -> "are" line 11 p.7: "higher-order" -> "high-order"?
>
> $\textbf{Response: }$
> 1) On "d can be larger than the ground truth", here we mean that the $d$ in formulation (5) and Theorem 1 can be larger than the $d$ in Assumption 1 because we usually do not know the true factorization size.  In the revision, we state
> ``...theoretically, which means the $d$ in formulation (5) and Theorem 1 can be larger than the $d$ (ground truth) in Assumption 1".
>
> 2) We corrected the typos and the grammar. Thank you again for the very nice comments, which improved our paper a lot.

---

### Official Review · Reviewer_jdoi · 2021-11-02

**Correctness:** 3
**Technical Novelty And Significance:** 3
**Empirical Novelty And Significance:** 2
**Recommendation:** 6
**Confidence:** 4

**Main Review:**

Overall, this is an interesting paper, providing an important theoretical analysis of deep matrix/tensor factorizations and bridging the gap between deep learning and tensor factorization models. The paper is well written and easy to follow; the theoretical analyses are delivered clearly despite the heavy notations. The two-mode design in the proposed matrix factorization is quite unique, to the best of my knowledge. Theoretical analysis and synthetic experiments show that when data is generated according to the assumption, it outperforms baselines and achieves better generalization. The idea of incorporating deep matrix factorization into Tucker decomposition is also quite interesting.

My questions/comments are:

1) In Assumption 2, it is assumed that X is generated by $\mathbf{X}=\mathbf{U}^\top\mathbf{Q}\mathbf{V}$, I wonder what motives this assumption and are there any motivating examples in the real world to justify this assumption?

2) The sizes of the tensors used for experiments are quite small, how would the proposed model scale to a large tensor? Also, the running time of the proposed model and the baselines are not reported.

3) There are other recent nonlinear tensor completion models, e.g. CoSTCo[1]. It would be great to see relevant discussion and comparison with those recent methods.


[1] Liu, Hanpeng, et al. "Costco: A neural tensor completion model for sparse tensors." Proceedings of the 25th ACM SIGKDD International Conference on Knowledge Discovery & Data Mining. 2019.



**Summary Of The Paper:**

This paper studies the nonlinear low-rank completion of matrices and tensors. Specifically, it first presents the theoretical results showing why nonlinear deep matrix factorization is better than the ordinary matrix factorization model. Then, it proposes a model named two-mode nonlinear deep matrix factorization to make full use of the nonlinearity of the nearly square matrices. The authors also extend this method to tensor factorization by further factorizing the factor matrices in the Tucker decomposition using the deep factorization method. Impressive results are obtained using both synthetic and real-world datasets.

**Summary Of The Review:**

This well-written paper presents some interesting and important theoretical results and proposes a promising multi-mode deep matrix and tensor factorization model. Its theoretical aspect seems solid to me, yet some of the experimental settings need to be clarified and evaluation against missing baselines is needed.

---

> ### Author Response · Authors · 2021-11-19
> **Detailed response**
>
> $\textbf{Q1:}$ In Assumption 2, it is assumed that $\mathbf{X}$ is generated by $\mathbf{X}=\mathbf{U}^\top\mathbf{Q}\mathbf{V}$, I wonder what motives this assumption and are there any motivating examples in the real world to justify this assumption?
>
> $\textbf{Response: }$ The assumption arose from that the data in some scenarios are generated by the linear interaction of two or more factors while the factors have low-dimensional nonlinear latent structures.  For example,  a covariance matrix $\mathbf{X}$ may be constructed naturally by data with nonlinear latent structures (e.g. $\mathbf{S}=f(\mathbf{Z})$), which matches our assumption when $\mathbf{U}=\mathbf{V}=\mathbf{S}$ and $\mathbf{Q}=\mathbf{I}$ or $\mathbf{U}\mathbf{Q}^{1/2}\triangleq\mathbf{S}$.  Similarly, one may imagine that the rating given by a user on an item is an interaction of the user’s features and item’s features and the features may have some nonlinear low-dimensional latent structures.
>
> In Figure 10 of the revised paper, we can see the nonlinearity in $\mathbf{U}$ intuitively. That's why our method outperformed the baseline methods significantly.  An intuitive visualization is provided in Figure 9. The superiority of our methods over the baselines stems from the deep nonlinear factorization and our methods outperformed the baselines significantly in most cases, which indicates that the assumptions are reasonable and effective in real tasks.
>
>
> $\textbf{Q2:}$ The sizes of the tensors used for experiments are quite small, how would the proposed model scale to a large tensor? Also, the running time of the proposed model and the baselines are not reported.
>
> $\textbf{Response: } $
> In our experiments, the sizes of the tensors are not large.  The main reason is that baseline methods TenALS, KBR-TC, TRLRF,  OITNN,  and our methods have high time costs on large tensors while all  codes were purely written in MATLAB and we do not have GPU to accelerate the computation.
>
> We added the computational complexity analysis in Appendix A of the revised paper.  The time cost in each iteration of the gradient-based optimization is $\mathcal{O}\left(kn^km+kn^{k-1}m^k \right)$ under some mild assumptions.
> The space complexity is $\mathcal{O}\big(\vert\Omega\vert+m^k+kn\sum_{l=1}^Lh_l\big)$.  Here $k$ is the order of the tensor,  $n$ is the side length of the tensor, and $m$ is the side length of the core tensor.  Now we see that the time (per iteration in the optimization) and space complexity of our M$^2$DMTF are slightly higher than the Tucker decomposition based tensor completion methods but much lower than KBR-TC(requires SVD) and TRLRF (tensor ring is costly).  Our M$^2$DMTF requires much more iterations than the baseline methods because it is composed of tensor decomposition and neural networks.
>
> In the revised paper, we reported the time costs (tensor completion on synthetic data) in Table 4. We also show them in the following table, where CoSTCo was performed in Python while all other methods were performed in MATLAB..  The time cost of our method is the highest because the iteration number is 3000. In contrast, the iteration numbers of other methods is about 500 or less (satisfied with the convergence conditions).
> $$
> \begin{matrix}
> Method  &FaLRTC & TenALS &TMac &KBR-TC & TRLRF &CoSTCo &OITNN-O &M^2DMTF \\\\ \hline
> Time (second) &3.5&	2.1&	1.4&	4.7&	4.3& 7.2&	6.8	&32.3\\\\
> \end{matrix}
> $$
>
> $\textbf{Q3:}$ There are other recent nonlinear tensor completion models,  e.g. CoSTCo[1].  It would be great to see relevant discussion and comparison with those recent methods.
>
> $\textbf{Response: }$ Thanks for the suggestion. It is a good baseline. We have included it in the revised paper. The comparison was added to Figure 4(b),  Figure 8(b), Table 7, and Figure 9.  In Figure 4(b), Figure 8(b), and Figure 9, the performance of CoSTCo (we used codes released by the authors) is not very good compared to other methods, although we have sufficiently tuned the hyper-parameters. For example,   the learning rate $lr$ was chosen from  $\{0.01, 0.001,0.0001\}$, the $rank$ was chosen  from $\{2,3,5,10,20\}$, the epoch number was 50 or 200, and the batch size was $256\times 2$ or $256\times 5$. We also tried to normalize the data to have zero mean and unit variance, or into the range of $[0,1]$ or $[-1,1]$ but the performance did not increase.  In Table 7 (the Movielens100k tensor),  CoSTCo outperformed TMac, TenALS, and OITNN. Our method M$^2$DMTF outperformed CoSTCo in all cases.
> $$
> \begin{matrix}\hline
> Missing~rate   &FaLRTC& TenALS &TMac &KBR-TC & TRLRF &CoSTCo &OITNN-O &M^2DMTF \\\\ \hline
> 0.5 & 0.2627 	&0.2813	&0.3065	 &0.2528	&0.2634	&0.2696	&0.2830	&\textbf{0.2429}\\\\
> 0.7 & 0.2692	&0.3097	&0.3470	&0.2685	&0.2719	&0.2724	&0.2951	&\textbf{0.2483}\\\\
> 0.9 &0.2881 	&0.3356	 &0.3782	&0.2739&0.2804	&0.2763	&0.3207	&\textbf{0.2614}\\\\ \hline
> \end{matrix}
> $$

---

> > ### Comment · Reviewer_jdoi · 2021-12-05
> > **Comments to authors' response**
> >
> > Sorry for the late reply and thank the authors for the detailed response as well as the added experiments. Most of my concerns are resolved, yet the added experiments do show that the proposed method is quite difficult to scale to large datasets. Especially when we compare with the best performing baseline, KBR-TC. The differences between them are not significantly large, yet the proposed model has a much slower running time. I would suggest the authors explore if the time complexity can be reduced in future works. Having said that, I will keep my rating of borderline accept unchanged because of the interesting ideas introduced in the paper.

---

> > > ### Author Response · Authors · 2021-12-05
> > > **Clarification about the time costs**
> > >
> > > Dear Reviewer,
> > >
> > > Thank you very much for your response and positive comment on our work. Indeed in the above table (Table 4 in the paper), the time cost of our method is almost 7 times of KBR-TC. The reason is that for the synthetic data, the maximum iteration of our method is 3000 while the stop criterial (relative change of the objective less than $10^{-5}$) never reached. As shown in Figure 7 the convergence of the objective function is a little slow but the recovery error converges faster. As analyzed in Appendix A, the per-iteration time complexity is slightly higher than Tucker-decomposition based tensor completion (e.g. TMac), which further indicates that the time cost per-iteration of our method is lower than KBR-TC (require SVD) , TRLRF, and OITNN-O (tensor ring is costly). Here we show recovery error and time cost of our method with different iterations in the case of synthetic tensor with missing rate 0.98.
> > > $$\begin{matrix}
> > >   &FaLRTC & TenALS &TMac &KBR-TC & TRLRF &CoSTCo &OITNN-O  \\\\ \hline
> > >  recovery\ error& 0.944&	0.973	&0.323&	0.169&	0.621&0.617&0.328 \\\\
> > > time\ cost\ (s)&3.5&	2.1&	1.4&	4.7&	4.3& \underline{7.2}&	6.8	\\\\
> > > \hline \hline
> > > & & & & & & &\\\\
> > > & &M^2DMTF\ &with\ &different\& iteration\& number\\\\ \hline
> > > iterations&100&200&300&500&1000&2000&3000\\\\
> > > recovery\ error&0.098&0.068&0.057&0.043&0.037&0.033&0.031\\\\
> > > time\ cost\ (s)&1.1&2.0&3.1&4.9&9.9&20.6&32.3\\\\
> > > \hline
> > > \end{matrix}
> > > $$
> > > We see that the recovery errors of all baselines are higher than 16% but the recovery error of our method is always less than 10\% and is less than 4% when the iteration number is 1000. Our method can outperform all baselines even when the iteration number is 100. As the max iterations of all baselines in this case are 500, we conclude that our method is as efficient as KBR-TC, TRLRF, CoSTCo, and OITNN-O.
> > >
> > > We also show the performance on the MovieLens100k tensor (a quite large tensor to our computational platform) below, where the missing rate is 0.7 and the max iterations of the baselines are 300.
> > > $$\begin{matrix}
> > > &KBR-TC & TRLRF &OITNN-O  \\\\ \hline
> > > recovery\ error  & 	0.2685	&0.2719	&0.2951\\\\ \hline
> > > time\ cost\ (s)& 754&283&966\\\\ \hline \hline
> > > & & & & \\\\
> > > &M^2DMTF\ &with\ different\& iteration\ number \\\\ \hline
> > > iteration &300 &1000&2000\\\\
> > > recovery\ error& 0.2494&0.2486&0.2483\\\\
> > > time\ cost\ (s)&227&771&1608\\\\
> > > \hline
> > > \end{matrix}
> > > $$
> > > We see that with the same max iteration, our method still outperformed the baselines on real data. The time cost of our method is less than KBR-TC and OITNN-O even when the iteration number of our method is 1000.
> > >
> > > These results verify that our method is more efficient and accurate than KBR-TC, TRLRF, and OITNN-O, which is consistent with our theoretical analysis. In future work, we would like to improve the convergence speed of the optimization and try much larger datasets. We hope that this clarification could address your concern about the scalability of our method. Thank you again.
> > >
> > > Sincererly,
> > >
> > > Authors.

---

### Official Review · Reviewer_26KD · 2021-11-08

**Correctness:** 4
**Technical Novelty And Significance:** 3
**Empirical Novelty And Significance:** 2
**Recommendation:** 5
**Confidence:** 4

**Main Review:**

Strengths:
(1) Strong theoretical analysis
(2) Clearly stating the assumptions in the proposed models

Weaknesses:
(1) Optimization procedure for the proposed models is not ver clear
(2) The applicability of the proposed models are evaluated only on matrix and tensor completion tasks which may raise doubts on usefulness of them on other downstream tasks.
(3) The experimental evaluation is not rich and strong. It is better to provide analysis on latent factors (output of matrix and tensor factorization). Besides, only a few baselines are included in the paper. More recent methods can be included as baselines for experimental evaluation. In addition, the proposed models do not consistently outperform the existing solutions on matrix and tensor completion tasks.
(4) Computational complexity analysis has not been performed elegantly for the proposed models.



**Summary Of The Paper:**

To bridge the gap between deep learning and tensor decomposition, this paper presents two novel approaches named as two mode non-linear deep matrix factorization and multi-mode nonlinear deep tensor factorization (extension of two mode model to multi-mode scenario). The main contribution of the methods lie in full exploration of non-linearity of data in matrix and tensor factorization. To better motivate the proposed models, the authors provide theoretical analysis for why and when nonlinear deep matrix factorization outperforms linear deep matrix factorization in matrix completion. The experimental evaluation demonstrates that in some datasets, the proposed models outperform the existing models on matrix and tensor completion tasks.

**Summary Of The Review:**

The proposed models built on top of meaningful assumptions for matrix and tensor factorization tasks which aim to address the gap between deep learning and factorization. However, there are several aspects in this paper that require improvement:

(1) The experimental evaluation is not very strong:
(1-1)I recommend the authors to provide in depth-analysis of the latent factors using visualization methods for downstream task of matrix and tensor completion.
(1-2)The authors need to explain why the proposed models do not achieve the best performance on some datasets.
(1-3)Please include more baselines for experimental evaluation
(2) Please include more details on the optimization procedures in the paper or appendix.
(3) I suggest the authors to put computational complexity analysis of the proposed models in the paper.

---

### Official Review · Reviewer_8BDW · 2021-11-09

**Correctness:** 4
**Technical Novelty And Significance:** 4
**Empirical Novelty And Significance:** 4
**Recommendation:** 6
**Confidence:** 4

**Main Review:**

The paper analyzed the proposed non-linear multi-mode tensor factorization algorithms, and the theoretical results help to understand the gap between the linear and non-linear factorization gap.

Some concerns are
[1] When comparing equation (6) and equation (7), some assumption regarding h seems missing. it is known that h_{L} = n1 and h_{-1}=n2, are there any assumptions needed  regarding the setups of h_{i: i = 1,..., L}? Are they bounded by r?

[2] There are some concerns about the synthetic data simulation used in the paper. It is okay to use complexed non-linear transformation (e.g. sin/exp) to verify the non-linear multi-mode tensor factorization works as expected. While, it would be better if the synthetic data can are generated the same way as the assumption (tanh activation function as used in the experiment) to verify how efficient of the algorithm in recovering a tensor with known factorization.


**Summary Of The Paper:**

This paper summarizes the theoretical guarantee for the existing LRMC and LRTC algorithms, and provides the theoretical analysis for a new proposed Multi-Mode Nonlinear deep tensor factorization. The analytical results show that when n2 is larger than n1, the nonlinear DMF provides a tighter generalization bound than MF. Similar analysis has been extended to two-mode matrix factorization and multi-mode  tensor factorization. Experimental results in synthetic data and real data show better results of the proposed algorithm as compared to other algorithms in completion tasks.

**Summary Of The Review:**

Overall, the paper is a theoretical paper and the bound analysis helps to understand the gap between classical linear factorization and non-linear factorization.

---

> ### Author Response · Authors · 2021-11-19
> **Detailed response**
>
> $\textbf{Q1:}$ When comparing equation (6) and equation (7), some assumption regarding h seems missing.  It is known that $h_{L} = n_1$ and $h_{-1}=n_2$, are there any assumptions needed regarding the setups of $h_{i: i = 1,..., L}$? Are they bounded by $r$?
>
> $\textbf{Response: }$ Thanks for pointing out.  In fact, they are not necessarily bounded by $r$ because we have assumed that $n_2$ is much larger than $n_1$. Then we only need that $\sum_{l=1}^{L-1}h_{l}h_{l-1}$ is not too large, which is realistic because we usually let $h_1,\ldots, h_{L-1}<n_1$.  In the revised paper, we modified the formulation as
>
> "Now comparing  (7) with (6), we conclude that the nonlinear DMF provides a tighter generalization bound than the classical MF when $n_2$ is much larger than $n_1$ and $\sum_{l=1}^{L-1}h_{l}h_{l-1}$ is not too large. The second condition is realistic because we usually let  $h_1,\ldots, h_{L-1}<n_1$ or we can just assume $h_1,\ldots, h_{L-1}<r$."
>
> $\textbf{Q2:}$ There are some concerns about the synthetic data simulation used in the paper. It is okay to use complexed non-linear transformation (e.g. sin/exp) to verify the non-linear multi-mode tensor factorization works as expected. While, it would be better if the synthetic data are generated the same way as the assumption (tanh activation function as used in the experiment) to verify how efficient of the algorithm in recovering a tensor with known factorization.
>
> $\textbf{Response: }$ Thanks for your nice suggestion. In the revised paper, we have added two experiments (one for synthetic matrix, the other for synthetic tensor),  in which we generated the data according to your suggestion, i.e.,  replaced the $\sigma, sin, cos, exp$ with $tanh$.  The results are reported in Figure 8 of the appendix. The comparative performances are similar to those in Figure 4 of the main paper.  It means that whether the data generating model are close to or quite different from the network structure we used does not matter.  The performances in Figure 4 are as good as Figure 8 because of the universal approximation ability of neural networks. The recovery errors of our methods in Figure 8 are not very small or close to zero because the solutions are not optimal due the difficulty in the optimization, though the data generating model is the same as the network structure.

---

### Author Response · Authors · 2021-11-19
**General response**

Dear Reviewers,

Thanks for your comments and time.  We have revised our paper according to the comments and suggestions.  The major modifications are summarized as follows.

1.  As suggested by Reviewer 26KD and Reviewer jdoi, we added more baselines and more experiments. Specifically, the methods of the following three papers have been included in the comparison studies. Then there are total 14 baselines including 7 matrix methods and 7 tensor methods. Most of them were proposed in recent 3 years.

[1] Yao Hu, Debing Zhang, Jieping Ye, Xuelong Li, and Xiaofei He. Fast and accurate matrix completion via truncated nuclear norm regularization. IEEE Transactions on Pattern Analysis and Machine Intelligence, 35(9):2117–2130, 2013.

[2] Christian Kummerle and Juliane Sigl. Harmonic mean iteratively reweighted least squares for low-rank matrix recovery. The Journal of Machine Learning Research, 19(1):1815–1863, 2018.

[3] Hanpeng Liu, Yaguang Li, Michael Tsang, and Yan Liu. Costco: A neural tensor completion model
for sparse tensors. In Proceedings of the 25th ACM SIGKDD International Conference on Knowledge Discovery $\&$ Data Mining, pp. 324–334, 2019.


The comparisons are reported in Figure 4, Figure 8, Table 7, and Figure 9.

We added a new experiment of tensor completion on the MovieLens-100k dataset. The results are reported in Table 7 of Appendix B.2.4. Our method outperformed other methods.

We also added two groups of plots (Figure 9 and Figure 10 in Appendix B.2.5) to visualize the recovery performance and learned latent factors of our method intuitively.

2.  As suggested by Reviewer 26KD, we provided the optimization details in Algorithm 1 of Appendix A and analyzed the computational complexity of our methods in Appendix A.  Also as suggested by Reviewer jdoi, we reported the time costs of the methods in Table 4 of Appendix B.1.2. The time cost of our method is higher than other methods because our method is composed of tensor decomposition and multilayer neural networks and hence needs much more iterations than other methods. In fact, the time cost per iteration of our method is comparable to or even less than some of the baselines such as KBR-TC, TRLRF, and OITNN.

3. As suggested by Reviewer 8BDW and Reviewer BPJG,  we improved the notations and definitions and made the formulations much clearer.

---

### Decision · Program_Chairs · 2022-01-20

**Decision:**

Accept (Poster)

**Comment:**

The paper considers matrix and tensor factorization, and provides a bound on the excess risk which is an improved bound over the bounds for ordinary matrix factorization. The authors also show how to solve the model with standard gradient-based optimization algorithms, and present results showing good accuracy. The method can be a bit slow but this depends a bit on the number of iterations, and in general it achieves better accuracy in a similar amount of time to other baseline algorithms.

The reviewers raised a few points, such as jdoi noting the tensor experiments were for small tensors and should include the method Costco as well; other reviewers mentioned more methods as well.  The authors seemed to address most of these concerns in the rebuttal, adding more experiments and more details on timing.  26KD mentioned the optimization procedure was unclear, but the revision includes pseudocode in the appendix that clarifies.

Overall, the paper has both a theoretical and algorithmic contribution, and would be of interest to many ICLR readers.